# Rodent ultrasonic vocal interaction resolved with millimeter precision using hybrid beamforming

**Max L Sterling[1,2,3], Ruben Teunisse[1], Bernhard Englitz[1]\***

[1]Computational Neuroscience Lab, Donders Institute for Brain, Cognition and Behaviour, Radboud University Nijmegen, Nijmegen, Netherlands; [2]Visual Neuroscience Lab, Donders Institute for Brain, Cognition and Behaviour, Radboud University Nijmegen, Nijmegen, Netherlands; [3]Department of Human Genetics, Radboudumc, Donders Institute for Brain, Cognition and Behaviour, Radboud University Nijmegen, Nijmegen, Netherlands

**Abstract** Ultrasonic vocalizations (USVs) fulfill an important role in communication and navigation in many species. Because of their social and affective significance, rodent USVs are increasingly used as a behavioral measure in neurodevelopmental and neurolinguistic research. Reliably attributing USVs to their emitter during close interactions has emerged as a difficult, key challenge. If addressed, all subsequent analyses gain substantial confidence. We present a hybrid ultrasonic tracking system, Hybrid Vocalization Localizer (HyVL), that synergistically integrates a high-resolution acoustic camera with high-quality ultrasonic microphones. HyVL is the first to achieve millimeter precision (~3.4–4.8 mm, 91% assigned) in localizing USVs, ~3× better than other systems, approaching the physical limits (mouse snout ~10 mm). We analyze mouse courtship interactions and demonstrate that males and females vocalize in starkly different relative spatial positions, and that the fraction of female vocalizations has likely been overestimated previously due to imprecise localization. Further, we find that when two male mice interact with one female, one of the males takes a dominant role in the interaction both in terms of the vocalization rate and the location relative to the female. HyVL substantially improves the precision with which social communication between rodents can be studied. It is also affordable, open-source, easy to set up, can be integrated with existing setups, and reduces the required number of experiments and animals.

**\*For correspondence:**
englitz@science.ru.nl

**Competing interest:** The authors declare that no competing interests exist.

## Editor's evaluation

This study introduces a novel and important hybrid strategy for recording ultrasonic vocalizations by combining data from several high quality microphones with data from a dense array of less sensitive microphones. This method enables recordings to be made from pairs and trios of freely interacting mice and accurate localization of their point of origin to convincingly determine the identity of the caller for each vocalization. This technology opens the door to new experiments incorporating analysis of vocal communication into behavioral paradigms.

## Introduction

Ultrasonic vocalizations (USVs) fulfill an important role in animal ecology as means of communication or navigation in many rodents (*Mahrt et al., 2013*; *Brudzynski, 2021*; *Zaytseva et al., 2019*; *Volodin et al., 2022*; *Murrant et al., 2013*), bats (*Schnitzler et al., 2003*), frogs (*Feng et al., 2006*), cetaceans (*Mourlam and Orliac, 2017*), and even some primates (*Bakker and Langermans, 2018*; *Ramsier*

**eLife digest** Most animals – from insects to mammals – use vocal sounds to communicate with each other. But not all of these sounds are audible to humans. Frogs, mice and even some primates can produce noises that are ultrasonic, meaning their frequency is so high they cannot be detected by the human ear. These 'ultrasonic vocalizations' are used to relay a variety of signals, including distress, courtship and defense.

To understand the role ultrasonic vocalizations play in social interactions, it is important to work out which animal is responsible for emitting the sound. Current methods have a high error rate and often assign vocalizations to the wrong participant, especially if the animals are in close contact with each other. To solve this issue, Sterling et al. developed the hybrid vocalization localizer (HyVL), a system which detects ultrasonic sounds using two different types of microphones. The tool is then able to accurately locate where an ultrasonic vocalization is emitted from within a precision of millimeters.

Sterling et al. used their new system to study courtship interactions between two to three mice. The experiments revealed that female courtship vocalizations were substantially rarer than previously reported when two mice were interacting. When three mice were together (one female, two males), Sterling et al. found that one of the male mice typically dominated the conversation. This result was also reflected by the male mouse locating themselves anogenitally to the female, as males tend to vocalize more when in this position.

In neuroscience, researchers often measure ultrasonic vocalizations to monitor social interactions between rats and mice. HyVL could provide neuroscientists with a more affordable and easier to use platform for conducting these kinds of experiments, which are important for studying behavior and how the brain develops.

---

*et al., 2012*). In many of these species, USVs have been shown to be present innately and to have significance at multiple stages of life, from neonates (*Kikusui et al., 2011*) to adults (*Mahrt et al., 2013*), often with diverse functions as distress/alarm calls (*Kikusui et al., 2011*; *Litvin et al., 2007*), courtship signals (*Marconi et al., 2020*), territorial defense signals (*Rieger and Marler, 2018*), private communication (*Ramsier et al., 2012*), and echolocation (*Schnitzler et al., 2003*). USVs have been extensively studied in mice, where their communicative significance has been widely demonstrated by their influence on conspecific behavior (*Hammerschmidt et al., 2009*; *Pultorak et al., 2017*; *Chabout et al., 2015*; *Musolf et al., 2015*; *Sugimoto et al., 2011*; *Tschida et al., 2019*; also in line with observational studies; *Warren et al., 2020*; *Nicolakis et al., 2020*; *Rieger et al., 2021*; *Petric and Kalcounis-Rueppell, 2013*). USVs can be grouped into different types that are highly context-dependent (*Chabout et al., 2015*; *Musolf et al., 2015*; *Nicolakis et al., 2020*; *Chen et al., 2021*; *de Chaumont et al., 2021*; *Castellucci et al., 2018*; *Pultorak et al., 2018*; *Burke et al., 2018*; *Zala et al., 2017a*; *Mun et al., 2015*; *von Merten et al., 2014*; *Scattoni et al., 2009*; *Warren et al., 2021*; *Dou et al., 2018*; *Hoier et al., 2016*; *Chabout et al., 2012*), and USV syntax itself is predictive of USV sequence (*Hertz et al., 2020*). Taken together, the current literature suggests USVs convey affective and social information in different behavioral contexts. This is further supported by the modulatory effect that testosterone and oxytocin have on USV production (*Kikusui et al., 2021b*; *Kikusui et al., 2021a*; *Timonin et al., 2018*; *Pultorak et al., 2015*; *Guoynes and Marler, 2021*; *Tsuji et al., 2021*; *Tsuji et al., 2020*). Importantly, the neuronal circuitry underlying USVs has recently been identified and is being studied extensively (*Tschida et al., 2019*; *Chen et al., 2021*; *Michael et al., 2020*; *Gao et al., 2019*; *Tasaka et al., 2018*; *Fröhlich et al., 2017*; *Shepard et al., 2016*; *Arriaga and Jarvis, 2013*; *Fujita et al., 2012*; *Wang et al., 2008*).

Because of their social and affective significance and our growing mechanistic understanding, mouse USVs are increasingly being used as a behavioral measure in neurodevelopmental and neuro-linguistic translational research (*de Chaumont et al., 2021*; *von Merten et al., 2014*; *Fröhlich et al., 2017*; *Yang et al., 2021*; *Binder et al., 2021*; *Hepbasli et al., 2021*; *Agarwalla et al., 2020*; *Tsai et al., 2012*; *Hodges et al., 2017*). Their manipulation and precise measurement not only provide the basis for tackling many fundamental questions but also pave the way, via advanced animal models, for the discovery of essential, novel drug targets for many debilitating conditions such as autism-spectrum disorder (*Tsai et al., 2012*; *Silverman et al., 2010*), Parkinson's disease (*Ciucci et al.,*

2009), stroke-induced aphasia (*Palmateer et al., 2016*), epilepsy aphasia syndromes (*Erata et al., 2021*), progressive language disorders (*Menuet et al., 2011*), chronic pain (*Palazzo et al., 2008*), and depression/anxiety disorders (*Moskal and Burgdorf, 2018*), where ultrasonic vocalizations serve as a biomarker for animal well-being and normal development. Consequently, we expect the scientific importance of mouse USVs to continue to increase in the coming years, highlighting the necessity to advance the methods required for their study. In recent years, substantial advances have been made in USV detection (*Coffey et al., 2019*; *Fonseca et al., 2021*; *Zala et al., 2017b*; *Van Segbroeck et al., 2017*; *Chabout et al., 2017*), classification (*Coffey et al., 2019*; *Fonseca et al., 2021*; *Van Segbroeck et al., 2017*; *Ivanenko et al., 2020*), and localization (*Oliveira-Stahl et al., 2023*; *Heckman et al., 2017*; *Warren et al., 2018a*; *Neunuebel et al., 2015*).

Localization is of particular importance during social interactions, when most USVs are emitted and any meaningful analysis of USV properties rests on a reliable assignment of each USV to its emitter. This task is complex for multiple reasons: (i) most USVs are emitted at close range, (ii) social behavior often requires free movement of the animals, and (iii) USV production is invisible (*Chabout et al., 2012*; *Mahrt et al., 2016*). With reliable assignment, all subsequent analyses can be conducted with substantial confidence concerning each USV's emitter. Although USVs could in theory be classified and assigned based on their shape (*Marconi et al., 2020*; *Liu et al., 2003*; *Holy and Guo, 2005*; *Barnes et al., 2017*; *Musolf et al., 2010*), this approach will depend strongly on different behavioral contexts and strains. Recent advances in acoustic localization (*Heckman et al., 2017*; *Warren et al., 2018a*; *Neunuebel et al., 2015*) have improved the localization accuracy to 11–14 mm; however, close-up snout–snout interactions – which is when a large fraction of USVs are emitted – require an even higher precision.

We have developed an advanced localization system for USVs in which is a high-resolution 'acoustic camera' consisting of 64 ultrasound microphones with an array of four high-quality ultrasound microphones. Both systems can individually localize USVs but exhibit rather complementary patterns of localization errors. We fuse them into a hybrid system that exploits their respective advantages in sensitivity, detection, and localization accuracy. We achieve a median absolute localization error of 3.4–4.8 mm, translating to an assignment rate of ~91%. Compared to the previous state of the art (*Oliveira-Stahl et al., 2023*; *Warren et al., 2018a*), the accuracy represents a threefold improvement that halves the proportion of previously unassigned USVs. Given the physical dimensions of the mouse snout (ø ~10 mm), this likely approaches the physical limit of localizability for USVs. We successfully apply it to and analyze dyadic and triadic courtship interactions between male and female mice. The comparison of dyadic and triadic interactions is chosen here as courtship interactions in nature are naturally competitive and this comparison is therefore both scientifically relevant and can benefit from high-reliability assignment of USVs. We demonstrate that the fraction of female vocalizations has likely been overestimated in previous analyses due to a lack of precision in sound localization. Further, in the triadic recordings we find that in competitive male–male–female courtship, one male takes a dominant role, which shows in emitting most USVs and also positioning itself more closely to the female abdomen.

## Results

We analyzed courtship interactions of mice in dyadic and triadic pairings. The mice interacted on an elevated platform inside an anechoic booth (see *Figure 1A*, for details see 'Recording setup'). Each trial consisted of 8 min of free interaction while movements were tracked with a high-speed camera (see *Figure 1B*), and USVs were recorded with a hybrid acoustic system composed of four high-quality microphones (i.e., USM4) as well as a 64-channel microphone array (Cam64, often referred to as an acoustic camera; see *Figure 1C* for raw data samples, green and red dots mark the start and stop times of USVs).

Most USVs were emitted in close proximity in dyadic and triadic pairings (see *Figure 1D*). Reliably assigning most USVs to their emitter therefore requires a highly precise acousto-optical localization system. The presently developed Hybrid Vocalization Localizer (HyVL) system is the first to achieve sub-centimeter precision, that is, ~3.4–4.8 mm (see *Figure 2* for an overview). This accuracy on the acoustic side is achieved by combining the complementary strengths of the USM4 and Cam64 data. The Cam64 data is processed using acoustic beamforming (*Van Veen and Buckley, 1988*), which delivers highly precise estimates (median absolute errors [MAE] = ~4–5 mm), but is not

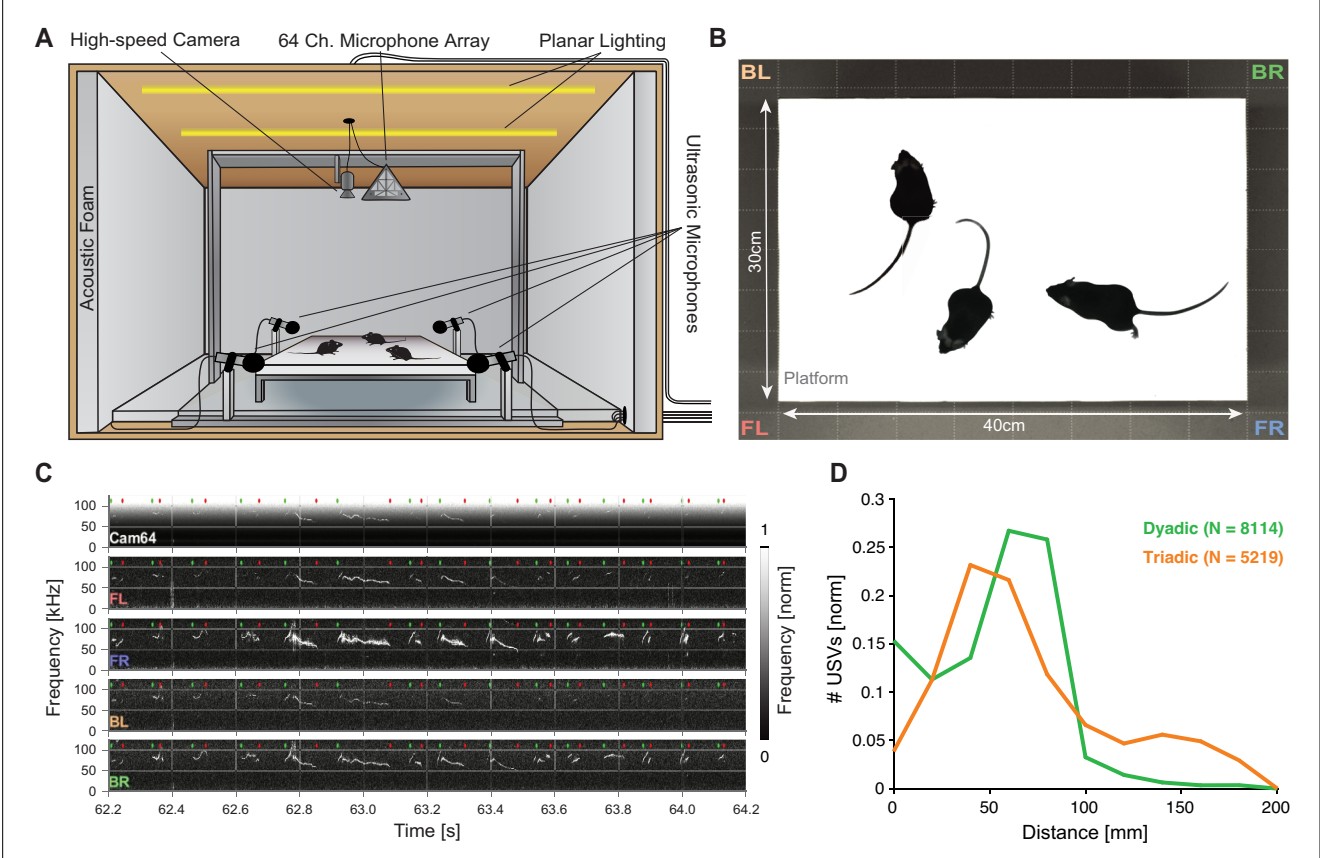

**Figure 1.** Mice emit ultrasonic vocalizations (USVs) in close proximity during courtship behavior. (**A**) Two or three mice of different sexes were allowed to interact freely on an elevated platform. Vocalizations were recorded with four high-quality ultrasonic microphones in a rectangular arrangement around the platform and a 64-channel microphone array ('Cam64') mounted above the platform. The spatial location of the pair was recorded visually with a high-speed camera. The platform was located in an ultrasonically sound-proof and anechoic box and illuminated uniformly using an array of LEDs. (**B**) Sample image from the camera that shows the high contrast between the mice and the interaction platform. The two-letter abbreviations indicate the locations of the four high-quality microphones (F = front, B = back, L = left, R = right). (**C**) Sample spectrograms from the four ultrasonic microphones and the average of all Cam64 microphones for a bout of vocalizations (start/end times marked by green/red dots). The Cam64 microphones are of lower quality than the USM4 microphones, evidenced by the rising noise floor for higher frequencies, affecting very-high-frequency USVs. (**D**) Most USVs in the present paradigm were emitted in close proximity to the interaction partners, with the vast majority within 10 cm snout–snout distance (i.e., ~93 and 72% for dyadic and triadic, respectively).

The online version of this article includes the following figure supplement(s) for figure 1:

**Figure supplement 1.** Comparison of noise spectra of the two microphone arrays.

sensitive enough for very-high-frequency USVs (see *Figure 1—figure supplement 1*). The USM4 data is analyzed using the previously published SLIM algorithm (*Oliveira-Stahl et al., 2023*), which delivers accurate (MAE = ~11–14 mm) and less frequency-limited estimates. The accuracy of SLIM, the previously most accurate ultrasonic localization technique (see 'Discussion' for a comparison), is generally lower than that of HyVL, but it makes essential contributions to the overall accuracy of HyVL through the integration of the complementary strength of the two methods/microphone arrays (see *Figure 3A and L*, shape of errors). The methods exhibit a complementary pattern of localization errors, which predestines them for high synergy when combined (see below).

For each USV, a choice is made between the USM4/SLIM and Cam64/Beamforming estimates based on a comparison of each method's USV-specific certainty and the relative position of the mice to the estimates, using an extended, hybrid Mouse Probability Index (MPI; *Neunuebel et al., 2015*). HyVL is the first system of its kind that exploits a hybrid microphone array to overcome the limitations of each subarray. The positions of the mice are obtained via manual and automatic video tracking using *DeepLabCut* (*Mathis et al., 2018*), each of which achieve millimeter precision for localizing the snout.

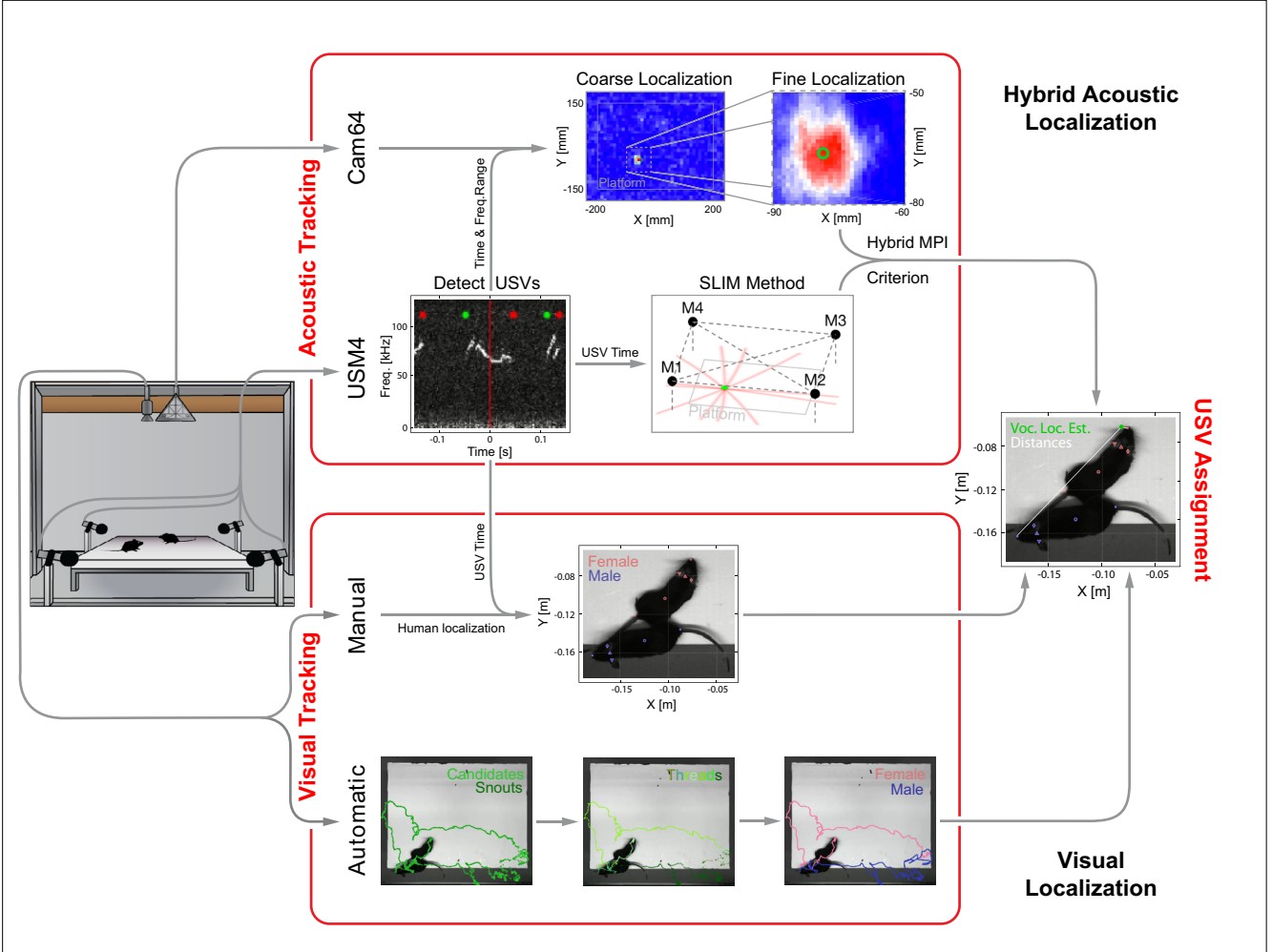

**Figure 2.** Overview of the combined acoustic and visual tracking pipeline. (Top) Acoustic tracking of animal vocalizations was enabled by a hybrid acoustic system, which recorded the sounds in the booth using a 64-channel ultrasonic microphone array ('Cam64') and four high-quality ultrasonic microphones ('USM4'). Vocalizations were automatically detected using USM4 data (start/end times marked by green/red dots) and then localized on the platform using both the SLIM algorithm on USM4 data and delay-and-sum beamforming on the corresponding Cam64 data. The Cam64 localization proceeded in two steps: first coarse (10 mm resolution), then fine centered around the coarse peak at 1 mm resolution (30 × 30 mm local window). The local, weighted average (green circle) was then used as the ultrasonic vocalization (USV) origin localized by Cam64. For each USV, the Cam64 localization was chosen if its SNR >5, otherwise the USM4/SLIM estimate was used (for details, see 'Localization of ultrasonic vocalizations'). (Bottom) Animals were tracked visually on the basis of concurrently acquired videos. Two tracking strategies were employed: (i) manual tracking in the video frames corresponding to the midpoint of USVs in all recordings and (ii) automatic tracking for all frames in dyadic recordings. (i) *Manual visual tracking*: the observer was presented with a combined display of the vocalization spectrogram and the concurrent video image at the temporal midpoint of each USV and annotated the snout and head center (i.e., midpoint between the ears). (ii) *Automatic visual tracking*: started with finding the optimal locations of each marker based on marker estimate clouds produced by *DeepLabCut* (*Mathis et al., 2018*) (DLC) for all frames. Next, these marker positions were assembled into spatiotemporal threads with the same, unknown identity based on a combination of spatial and temporal analysis. Finally, the thread ends still loose were connected based on quadratic spatial trajectory estimates for each marker, yielding the complete track for both mice (see 'Automatic visual animal tracking' and *Figure 3—figure supplement 1*).

Overall, 228 recordings were collected from 14 male and 4 female mice (153 dyadic, 67 triadic, and 8 with a single mouse). In 90 recordings, USVs were produced and recorded with Cam64 and USM4 simultaneously (55 dyadic, 28 triadic, and 7 single). The single mouse recordings were also used in a previous publication (*Oliveira-Stahl et al., 2023*) where only the SLIM accuracy was evaluated. A total of 112 recordings were recorded in a balanced design (four dyadic and four triadic per male mouse paired with all females) and the remaining recordings conducted with good vocalizers to maximize the number of USVs for downstream analysis. In all trials combined, 13714 USVs were detected.

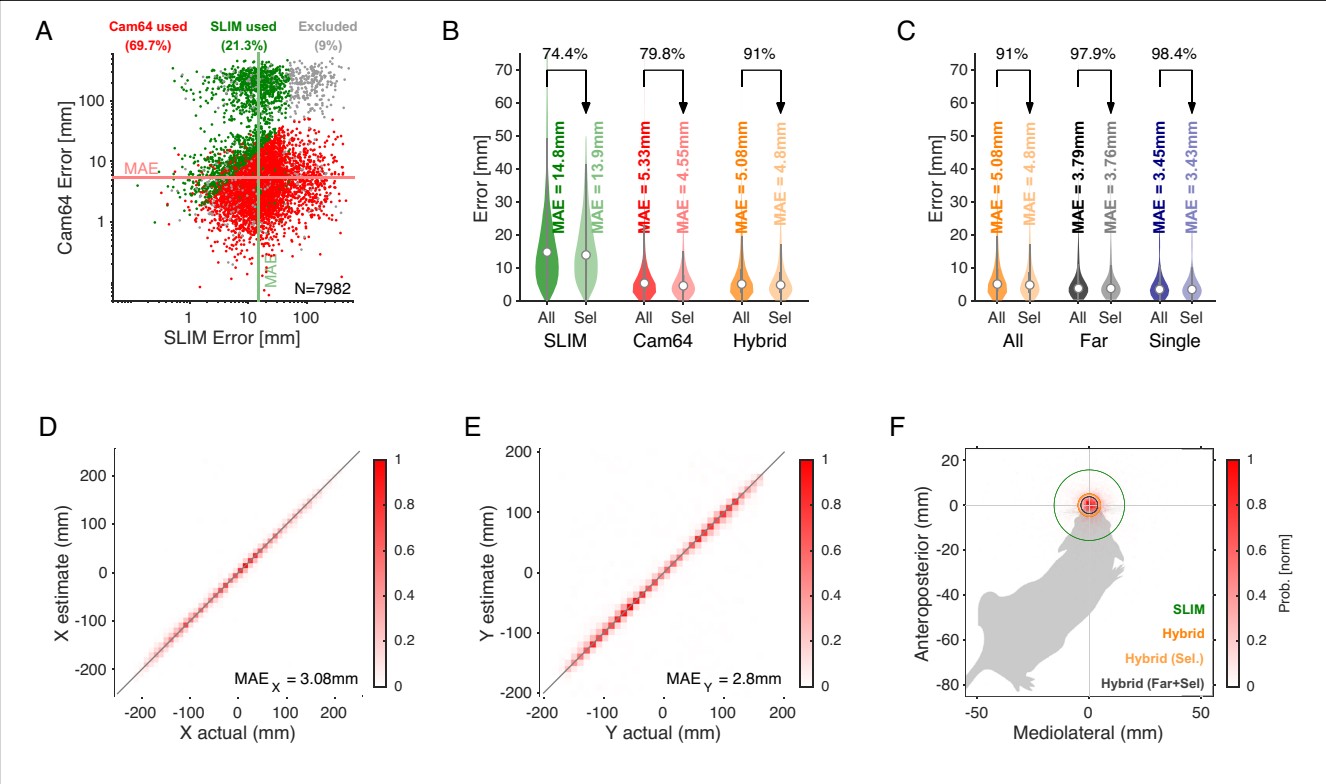

**Figure 3.** Spatial accuracy of localizing ultrasonic vocalizations (USVs) during mouse social interaction improves approximately threefold over the state of the art (*Oliveira-Stahl et al., 2023*). (**A**) The vast majority of USVs is localized with very small errors for both methods, concentrated close to the axes and thus hardly visible, evidenced by the median absolute errors (MAE) for Cam64 (light red line) and SLIM (light green line). The fewer larger errors form an L-shape, emphasizing the synergy of a hybrid approach that compensates for the weaknesses of each method. Location estimates were excluded (gray) if they were >50 mm from either mouse, or the hybrid Mouse Probability Index (MPI) <0.95. (**B**) The hybrid localization system Hybrid Vocalization Localizer (HyVL) (orange) combines the virtues of SLIM and Cam64 enabling the localization of 91.1% of all USVs (light orange), achieving an MAE = 4.8 mm. Cam64 localization (red) alone only includes 74.4% of all USVs, but at an MAE = 4.55 mm (light red). SLIM-based localization (green) only includes 79.8% of all USVs, at an MAE = 14.8 mm (light green, see 'USV assignment' for details on the relation between accuracy and selection criteria). (**C**) USVs emitted when all animals were >100 mm apart and a single mouse condition was used to assess the ideal accuracy of HyVL. For the far condition, virtually all USVs (332/339, 97.9%) were assigned at an MAE = 3.79 mm, similarly to the single animal condition (MAE = 3.45 mm, 251/255, 98.4%). (**D, E**) Comparison of actual with estimated snout locations along the X (horizontal; **D**) and Y (vertical; **E**) dimensions indicating strong agreement. Colors indicate peak-normalized occurrence rates. (**F**) Centered overlay of USV localizations relative to emitter snout. Precision is depicted as a circle with a radius equivalent to the median absolute error (green: SLIM; orange: HyVL, all USVs; light orange: HyVL, selected USVs, dark gray: HyVL, when mice >100 mm apart).

The online version of this article includes the following figure supplement(s) for figure 3:

**Figure supplement 1.** Schematic depiction of the progression of marker localization and identity attribution.

**Figure supplement 2.** Ground truth localization of band-limited noise emitted from a small speaker.

**Figure supplement 3.** Localization results on the basis of automatic tracking from dyadic and triadic recordings.

## Precision of USV localization

Assigning USVs to individual mice required combining high-speed video imaging with the HyVL location estimates at the times of vocalization. We manually tracked the animal snouts at the temporal midpoint of each USV to obtain near-optimal position estimates (see *Figure 2*). We first assessed the relative structure of the localization errors between both methods, USM4/SLIM (*Figure 3A*, green) and Cam64/Beamforming (*red*, each dot is a USV). While most errors were small, and clustered close to the origin of the graph (evidenced by the small MAE, shown as horizontal and vertical lines, respectively), the less frequent, larger errors exhibited an L-shape. This error pattern is an optimal situation for combining estimates from the two methods, to compensate for each other's limitations. While the Cam64 data can compensate for single microphone noise through the large number of microphones, the nature of its micro-electromechanical systems (MEMS) microphones deteriorates for very

high frequencies (see *Figure 1—figure supplement 1B*). Conversely, the USM4 microphones show an excellent noise level across frequencies (see *Figure 1—figure supplement 1A*) but can produce erroneous estimates if there is noise in a single microphone and have an intrinsic limitation in spatial accuracy due to the physical size of their receptive membrane (ø ~20 mm).

We therefore designed an analytical strategy to combine the estimates of both systems to optimize the number of reliably assignable USVs, while evaluating the resulting spatial accuracy alongside. Briefly, the location estimates of both methods each come with an estimate of localization uncertainty. First, we assess for each method's estimate how reliably it can be assigned to one of the mice, taking into account the positions of the other mice. This is quantified using the MPI (*Neunuebel et al., 2015*), which compares the probability of assignment to a particular mouse to the sum of probabilities for all mice, weighted by the estimate's uncertainty. If the largest MPI exceeds 0.95, it is considered a reliable assignment to the corresponding mouse. If both methods allowed reliable assignments, the one with smaller residual distance was chosen. If only one method was reliable for a particular USV, its estimate was used. If neither method allowed for reliable assignment, the USV was not used for further analysis. This typically happens if the snouts are extremely close or the USV is very quiet. This approach outperformed many other combination approaches in accuracy and assignment percentage, for example, maximum likelihood (see "Assigning USVs' and 'Discussion' for details).

Analyzing all courtship vocalizations, HyVL performed significantly better than either method alone (see *Figure 3*), allowing a total of 91.1% of USVs to be assigned at a spatial accuracy of 4.8 mm (MAE). This constitutes a substantial 2.9-fold improvement in accuracy over the previous state of the art, the SLIM algorithm (*Oliveira-Stahl et al., 2023*). On the full set of USVs where both microphone arrays were recording (N = 7982), HyVL outperformed both USM4/SLIM and Cam64/Beamforming significantly, both in residual error (SLIM: 14.8 mm; Cam64: 5.33 mm; HyVL: 5.08 mm; p<10⁻¹⁰ for all comparisons, Wilcoxon rank-sum test) and percentage of reliably assigned USVs (SLIM: 74.4%; Cam64: 79.8%; HyVL: 91.1%). Cam64/Beamforming performed even more precisely on its reliably assignable subset (4.55 mm), which was, however, smaller than the HyVL set. This difference emphasizes the complementarity of the two methods and thus the synergy through their combination. There was no significant difference between tracking on dyadic and triadic recordings (HyVL: 5.0 mm vs. 5.1 mm, p=0.71, Wilcoxon rank-sum test) with correspondingly similar selection percentages (92 vs. 90%, respectively).

The accuracies above are an average over localization performance at any distance. In particular during close interaction, USVs will often be reflected or obstructed, complicating localization. While this constitutes the realistic challenge during mouse social interactions, we also investigated the 'ideal', unobstructed performance of HyVL by comparing the performance on USVs emitted when all animals were 'far' (>100 mm) apart, that is, >~20 times the average accuracy of HyVL, as well as for a single male mouse on the platform. For the *far* USVs, the reliably assignable fraction increased to 97.9%, and the accuracy significantly improved to 3.79 mm (*Figure 3C* gray, p=8.6 × 10⁻⁷, Wilcoxon rank-sum test). For the *single animal* USVs, the accuracy was even better at 3.45 mm with 98.4% reliably assigned (*Figure 3C*, blue). In addition, we evaluated HyVL's performance on sounds emitted from a miniature speaker placed in a regular grid of locations (see *Figure 3—figure supplement 2*). In this condition, the accuracy was even higher (1.87 mm, or even ~0.5 mm when correcting for experimental factors, see figure caption); however, given the differences in the emitter characteristics, emitted sounds and lack of adsorption, this should be treated as a lower bound that will be hard to achieve with mice.

Next, we inspected separate localization along the X and Y axis to check for anisotropies of localization (*Figure 3D/E*, histograms normalized to maximum). The position of the closest animal aligned precisely with the estimated position in both dimensions, indicated by the high density along the diagonal (*Pearson r* > 0.99 for both dimensions) and the MAE's along the X and Y direction separately (X = 3.1 mm, Y = 2.8 mm). These one-dimensional accuracies might be of relevance for interactions where movement is restricted.

Lastly, we visualized the localization density relative to the mouse that the vocalization was assigned to (*Figure 3F*). Combining both dimensions and appropriately rotating them, the estimated position of the USVs is shown relative to the mouth. The density is narrowly centered on the snout of the mouse (circle radius = MAE: green: SLIM method; orange: HyVL; light orange: HyVL assigned USVs; gray: far assigned USVs).

In summary, the HyVL system provides a substantial improvement in the localization precision. In comparison to other methods, its precision also allows a larger fraction of vocalizations to be reliably assigned and retained for later analysis, which enables a near complete analysis of vocal communication between mice or other vocal animals (see 'Discussion' for details).

## Sex distribution of vocalizations during social interaction

Courtship interactions between mice lead to high rates of vocal production, but are challenging due to the relative proximity, including facial contact. Previous studies using a single microphone have often assumed that only the male mouse vocalized (*Rotschafer et al., 2012*; *Choi et al., 2011*; *Pomerantz and Clemens, 1981*; *Nunez et al., 1978*), while more recent research has concluded that female mice vocalize as well (*Neunuebel et al., 2015*; *Sangiamo et al., 2020*). Female vocalizations were typically less frequent, but constituted a substantial fraction of the vocalizations (11–18%) (*Oliveira-Stahl et al., 2023*; *Heckman et al., 2017*; *Neunuebel et al., 2015*; *Warren et al., 2018b*). Below, we demonstrate that the accuracy of the localization system can be an important factor for conclusions about the contribution of different sexes to the vocal interaction.

Over all dyadic and triadic trials combined, females produced the minority of vocalizations. Naive estimation without MPI selection using SLIM estimates ~14%, while HyVL tallies it at just 7% (*Figure 4A*). Applying MPI selection, SLIM estimates only 5.5%, while HyVL arrives at significantly less, just 4.4% (p=0.002, paired Wilcoxon signed-rank test, *Figure 4A/B*), while reliably classifying 91.1% of all vocalizations.

Using HyVL instead of SLIM significantly reduces the fraction of female vocalizations, suggesting that less accurate algorithms overestimate the female fraction (only results for MPI-selected USVs shown, *Figure 4B*). Considering only vocalizations that are emitted when the snouts are >50 mm apart further significantly reduces the fraction to female USVs to 1.1% after MPI selection (p=5.2 × $10^{-8}$, Wilcoxon rank-sum test). Comparing the percentage of female vocalizations between dyadic and triadic trials, no significant differences were found (p=0.22, Wilcoxon rank-sum test, *Figure 4D*).

Beyond the absolute distance between the mouths of the mice, high-accuracy localization of USVs allows one to position the bodies of the animals relative to one another at the times of vocalization by combining acoustic data with multiple concurrently tracked visual markers. This provides an occurrence density of other mice relative to the emitter (*Figure 4E*).

Female mice appear to emit vocalizations in very close snout–snout contact, with a small fraction of vocalizations occurring when the male snout is around the hind-paws/ano-genital region (*Figure 4F*). Male mice emit vocalizations both in snout–snout contact, but also at greater distances, which dominantly correspond to a close approach of the male's snout to the female ano-genital region (*Figure 4G*). This was verified separately with a corresponding analysis, where the recipient's tail-onset was used instead (not shown).

In summary, the combination of high-precision localization and selection using the MPI indicates that female vocalizations may be even less frequent than previously thought. When they vocalize, the mice appear to almost exclusively be in close snout–snout contact. As this is incidentally also the condition that has the highest chance of mis-assignments, even the remaining female vocalizations need to be treated with caution.

## Vocalization rate analysis

In dyadic trials, one female and one male mouse interacted, whereas in triadic trials either two males and one female or two females and one male mouse interacted. We first address in dyadic trials, whether there were significant differences in individual vocalization rates between the mice. For the balanced dataset of 14 × 4 dyadic interactions (pairing of all males with all females), we did not find a significant effect of individual on vocalization rates for either male and female mice (see *Figure 5—figure supplement 1*, p=0.46 and p=0.16, respectively, one-way ANOVA analysis with factor individual, for n = 4 recordings in males and n = 14 recordings in females). For triadic trials, we could not perform the corresponding analysis since the two male/female recordings could not be distinguished reliably in post hoc tracking.

In the balanced dyadic and triadic datasets, only 23/112 recordings contained vocalizations. We collected additional dyadic and triadic recordings for the purpose of maximizing the number of USVs, both for assessing HyVL performance and comparing dyadic and triadic interactions. In this enlarged

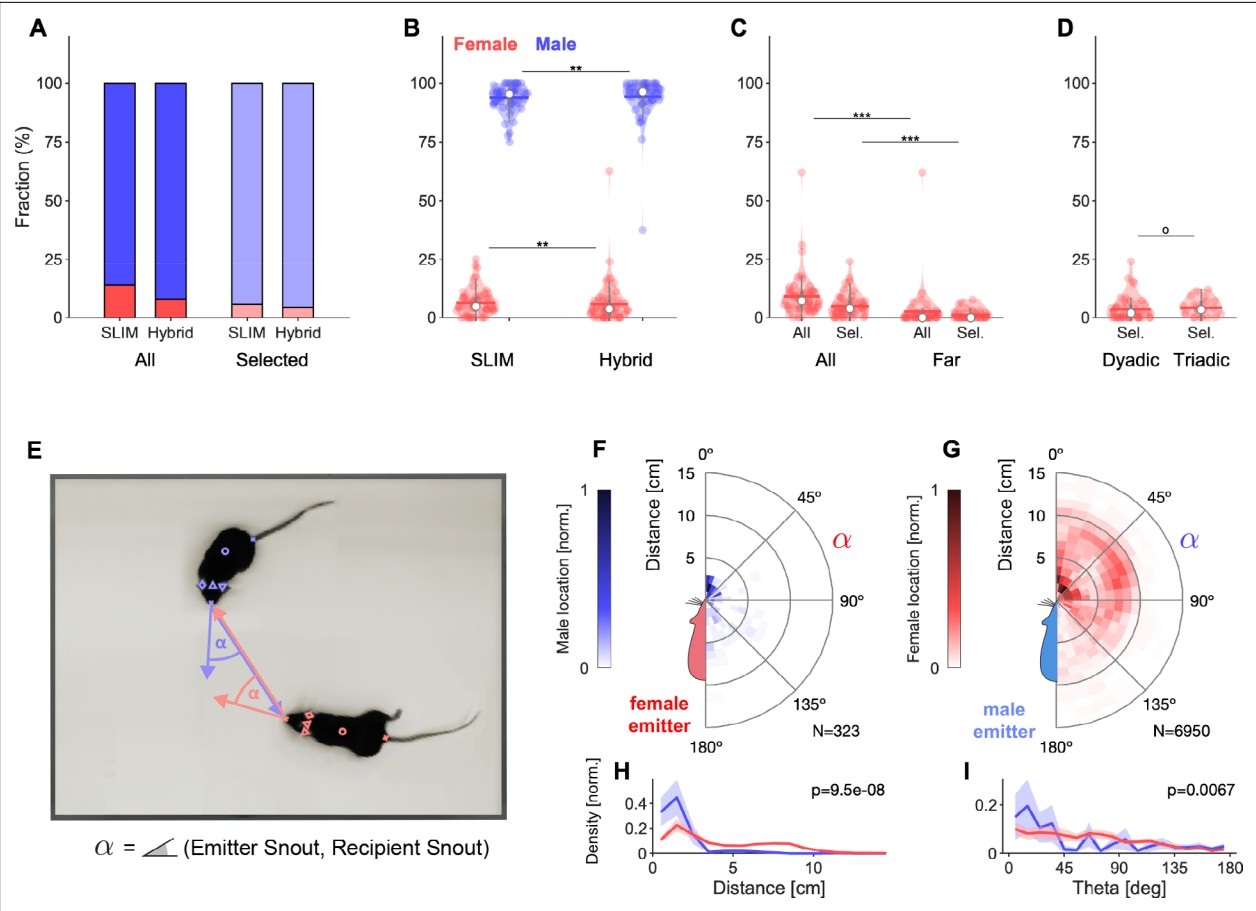

**Figure 4.** Analysis of sex-dependent vocalizations can depend on localization accuracy. (**A**) Female vocalizations constitute a small fraction of the total set of vocalizations. The female fraction further reduces with increased precision and when selecting vocalizations based on the Mouse Probability Index (MPI). Vocalization fractions are separated by sex, not by individual mouse. Fractions include all dyadic and triadic recordings with ultrasonic vocalizations (USVs) (N = 83), same for all other panels. (**B**) Using the hybrid method instead of SLIM significantly reduces the fraction of female vocalizations, suggesting that less accurate algorithms overestimate the female fraction (only results for MPI-selected USVs shown). (**C**) The fraction of female vocalizations further reduces if only USVs are considered that are emitted while all animal snouts were >50 mm apart from each other. This indicates a preference of female mice to vocalize in close snout–snout contact; however, this entails that female vocalizations are more prone to confusion with male vocalizations due to their relative spatial occurrence. (**D**) There was no difference in the female fraction of USVs between dyadic and triadic pairings (two male and two female conditions combined here; $N_{Dyadic}$ = 55, $N_{Triadic}$ = 28). (**E**) High-accuracy localization of USVs allows one to analyze the relative spatial vocalization preferences of the mice, that is, their occurrence density in relation to the relative position of other mice to the emitter. We quantified this by collecting the position of the nonvocalizing mice at the times of vocalization, in relation to the vocalizing mouse. Symbol α corresponds to the angle between the emitter's snout and the snout of other mice. (**F**) Female mice appear to emit vocalizations in very close snout–snout contact, with a small fraction of vocalizations also occurring when the male mouse around the hind-paws/ano-genital region. (**G**) Male mice emit vocalizations both in snout–snout contact, but also at greater distances, which dominantly correspond to a close approach of the male's snout to the female ano-genital region. This was verified separately with a corresponding analysis, where the recipient's tail-onset was used instead (not shown). (**H**) Radial distance density of receiver animals, marginalized over directions, shows a significant difference, with females vocalizing mostly when males (blue) are in close proximity of the snout, while males vocalize when the female mouse's snout is very close (corresponding to snout-snout contact), but also when the female's snout is about 1 body length away (snout–ano-genital interaction). Plots show means and SEM confidence bounds. (**I**) Direction density of receiver animals, marginalized over distances, shows that female mice vocalize primarily when the male mouse's snout is very close and in front of them. Note that the overall angle of approach of the male mouse is not from directly ahead (see *Figure 4—figure supplement 1*).

The online version of this article includes the following figure supplement(s) for figure 4:

**Figure supplement 1.** Relative spatial vocalization preferences relative to receiver's ano-genital region for dyadic recordings.

dataset, a total of 83 recordings (55 dyadic, 28 triadic) were available, which contained USVs. This dataset was still balanced for female mice, but, unbalanced for male mice, that is, although the same mice participated in both dyadic and triadic recordings, however, not with exactly the same number of recordings. While the analysis on the balanced dataset above did not suggest significant differences

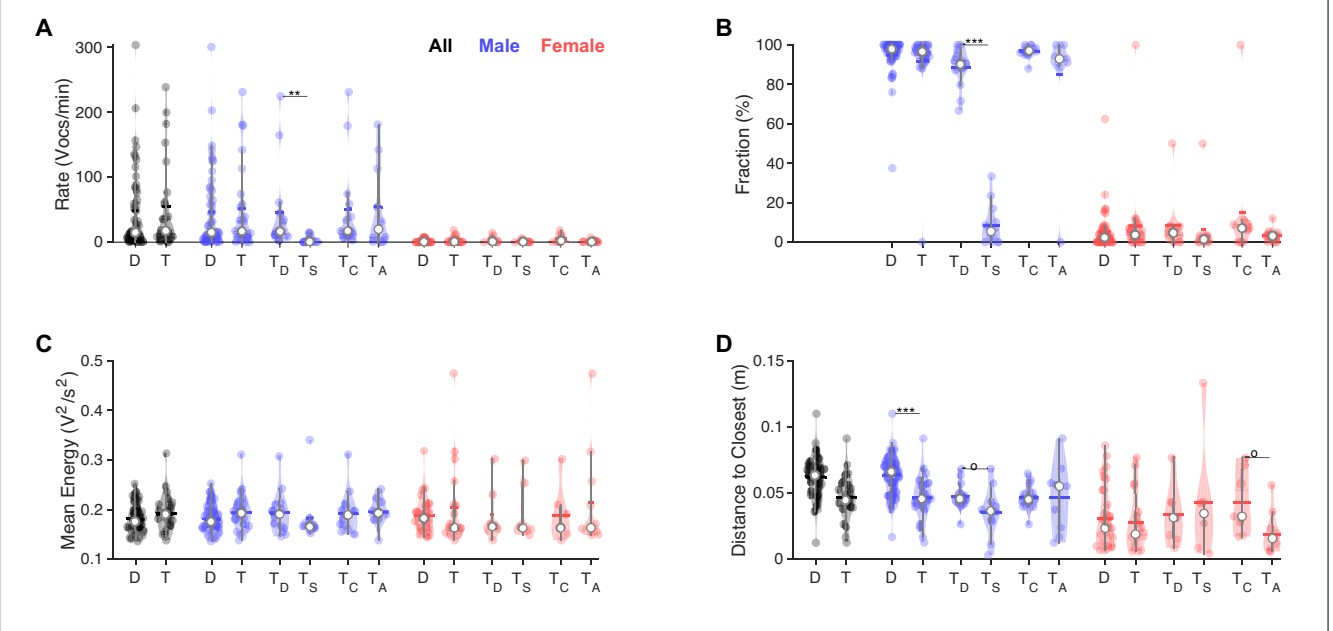

**Figure 5.** In triadic interaction, one male vocalizes dominantly and males vocalize even closer to females. (**A**) Overall, vocalization rates were comparable between dyadic (D) and triadic (T) conditions. Male mice (blue) vocalized at higher rates than female mice (red). However, this was restricted to the dominant male mouse ($T_D$: dominant = emitted more ultrasonic vocalizations [USVs] within same-sex) in triadic, competitive (2 m/1 f) conditions (see text for all p-values). Male vocalization rates were similar in competitive ($T_C$: with same-sex competitors) and alternative ($T_A$: no same-sex competitor, i.e., for male vocs: 2 f/1 m) pairings. Female vocalization rates remained low and similar across all conditions. $T_S$: submissive mouse = emitted fewer USVs within same sex during competitive trial; white dot: median; horizontal bar: mean (N = 83 recordings in all panels, in the groupings D/T vocalizations are grouped by sex, whereas in $T_{D,S,C,A}$ USVs are per individual, same in panels **B–D**). (**B**) While the fraction of USVs emitted by males was overall comparable between D and T pairings, the dominant male ($T_D$) emitted a substantially larger fraction than their submissive counterpart ($T_S$), roughly a factor of 9. In competitive pairings, male mice tended to emit an overall larger fraction of all USVs than in alternative pairings ($T_C$ vs. $T_A$), but this is unsurprising as both males vocalize. In female mice, the overall fraction of USVs in D and T pairings was also similar (see details in 'Results' for potential caveats of the dominant/subordinate classification). (**C**) In triadic pairings, dominant male mice tended to vocalize more intensely than in dyadic pairings; however, this difference was not significant at the current sample size. No significant differences were found for female mice. (**D**) Male mice emitted USVs in closer proximity to the closest female mouse in triadic compared to dyadic interactions. Female mice generally emitted USVs at closer distances (see also *Figure 4F/H*), in particular for alternative vs. competitive pairings.

The online version of this article includes the following figure supplement(s) for figure 5:

**Figure supplement 1.** Vocalization rates in dyadic recordings based on a balanced set of four recordings per male mouse and condition (n = 14 male mice, 112 recordings).

between individuals, we thus cannot fully exclude that the reported differences below are partially due to individual differences between some male mice.

In the analysis of triadic interactions, we separate competitive and alternative contexts depending on whether a mouse had to compete with another same sex mouse or could interact with two opposite sex mice, respectively. For triadic trials we further separate the same-sex mice into dominant and subordinate, based on who vocalized more.

However, in competitive interactions between males, one male mouse significantly and strongly dominated the 'conversation,' with on average ninefold more vocalizations than the other male mouse ($T_D$ vs. $T_S$, *Figure 5A and B*, both comparisons: p<0.005 [Wilcoxon sum of ranks test]) after Bonferroni correction. Specifically, Bonferroni correction was conducted per panel/measured variable on the basis of the number of hypotheses actually tested for, that is, six tests per panel, three for each sex: dyadic vs. triadic; triadic: dominant vs. subordinate; triadic: competitive vs. alternatives. While the present division into dominant and subordinate mouse based on a higher vocalization rate within a recording will always lead to a significant difference, the quantitative difference between them is the striking aspect in this comparison. Overall male vocalization rates were similar in competitive and alternative triadic trials. Female vocalization rates were similar across all compared conditions.

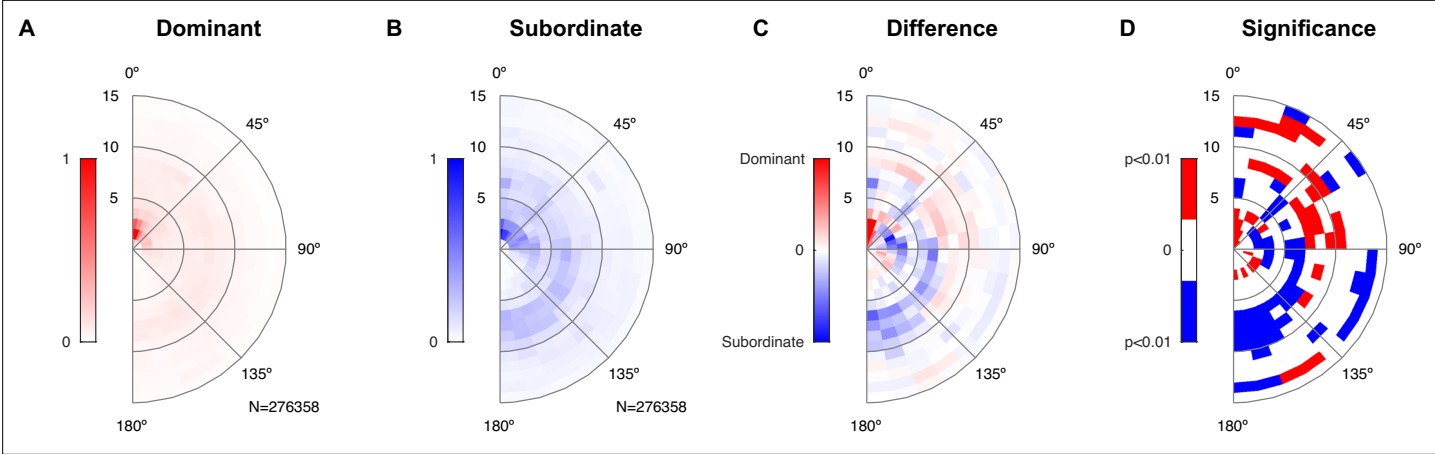

**Figure 6.** Dominant male animals spend more time close to the female's abdomen. (**A**) The abdomen of the female was typically close to the *dominant* male's snout (center of plot), with a ring of approximately one mouse length also visible deriving from snout–snout interactions. The histogram was created based on all-frame tracking of the 14 triadic interactions with two male mice using skeleton tracking in SLEAP over a total of N = 276,358 frames. Dominant and subordinate males were defined based on their vocalization rate per recording. Each histogram was peak normalized. (**B**) For the *subordinate* male, the histogram was less peaked around the proximal snout–abdomen interactions, but showed a more visible arc between 90 and 180°, pointing to snout–snout interactions. (**C**) The difference between the two histograms (each density-normalized to a sum of 1) shows the focused snout–abdominal interactions for the dominant male, and the arc pointing to snout–snout interactions for the subordinate male, in addition to smaller absolute differences in other relative locations. (**D**) Spatial regions of significant difference between the dominant and subordinate male were found both in the regions highlighted in (**C**), as well as more distant regions. Significance was assessed by bootstrapping confidence bounds on the histograms of the dominant and subordinate males (based on relative locations, rebuilding the histogram, 100×). The distance to the most extreme values were taken as the limits for significant deviation at p<0.01, and the difference in (**C**) was then compared in both the positive/negative direction against these bounds.

The mean vocalization energy of dominant males in triadic pairings tends to be higher than those of submissive males in triadic pairings; however, this result did not reach significance in the present dataset (see *Figure 5C*). No effects of vocalization energy were found in females.

The distance to the closest animal of the *opposite sex* was found to be even closer during triadic trials (see *Figure 5D*), driven purely by male vocalizers (p=0.00046, after Bonferroni correction as above, Wilcoxon sum of ranks test): the distance to the closest animal does not change between conditions for vocalizing females (p=0.975, Wilcoxon sum of ranks test). Interestingly, the distance to the closest animal was larger for females at the time of vocalization when they had a same-sex competitor on the interaction platform with them than when they were the only female ($T_c$ vs. $T_a$, p=0.0068, Wilcoxon sum of ranks test).

Lastly, we investigated whether the division into a dominant and subordinate male based on the vocalization rate was also reflected in the spatial behavior of the male mice relative to the female mouse. For this purpose, we again constructed relative spatial interactions histograms (see *Figure 6*, analogous to *Figure 4*), separately for USV-rate-dominant and subordinate males. The results are displayed as the relative location between the male snout and the female abdomen. Dominant males spent more time close to the female abdomen, thus engaging in ano-genital contact (*Figure 6A*, center), in comparison with subordinate males (*Figure 6B*). This is highlighted in the difference between the spatial interaction histograms (*Figure 6C*), where the most salient dominant peak occurs in the center, while the subordinate male spent more time in snout–snout contact, indicated by the blue arc at about one mouse body length from the center (shown in blue here). These differences were significant, in addition to a number of other locations in the spatial interaction histogram. Significance analysis was performed using 100× bootstrapping on the relative spatial positions to estimate p=0.99 confidence bounds around the histograms of the dominant and subordinate, respectively. Significance at a level of p<0.01 highlights multiple relative spatial positions.

In summary, in competitive triadic interactions, one of the male mice took a strongly dominant role, evidenced both in the vocalization rate and the more abundant ano-genital interactions with the female throughout the recordings. In triadic interactions, the female mouse was generally approached more closely by a male mouse, in particular in the alternative condition. The latter could, however, be

a consequence of the larger number of male animals on the platform compared to dyadic and triadic competitive (from the perspective of the female).

## Discussion

We have developed and evaluated a novel, hybrid sound localization system (HyVL) for USVs emitted by mice and other rodents. USVs are innately used by rodents to communicate social and affective information and are increasingly being used in neuroscience as a behavioral measure in neurodevelopmental and neurolinguistic research. In the context of dyadic and triadic social interactions between mice, we demonstrate that HyVL achieves a groundbreaking increase in localization accuracy down to ~3.4–4.8 mm, enabling the reliable assignment of >90% of all USVs to their emitter. Further, we demonstrate that this can be combined with automatic tracking, enabling a near-complete and automated analysis of vocal interaction between rodents. The showcased analyses demonstrate the advantages obtained through more precise localization, further discussed below. HyVL is based on an array of high-quality microphones in combination with a commercially available, affordable acoustic camera. With our freely available code, this system can be readily reproduced by other researchers and has the potential to revolutionize the study of natural interactions of mice.

### Comparison with previous approaches for localizing vocalizations

Localization accuracy was first systematically reported by *Neunuebel et al., 2015* using a four-microphone setup and a maximum likelihood approach (*Zhang et al., 2008*), who attained an MAE of ~38 mm that conferred an assignment rate of 14.6–18.1% (their Table 1, *assigned* relative to *detected* or *localized*). Originating from the same research group, *Warren et al., 2018a* employed both a four- and eight-microphone setup in a follow-up study, achieving an MAE of ~30 mm for four microphones (~52% assignment rate) and ~20 mm with eight microphones (~62% assignment rate), both using a jackknife approach to increase robustness of localization. *Oliveira-Stahl et al., 2023* introduced the SLIM algorithm, reaching an MAE of ~11–14 mm (~80–85% assignment rate depending on the dataset) using four microphones. Presently, we advance the state of the art in multiple ways: we use 68 microphones, combining a 64-channel 'acoustic camera' with four high-quality ultrasonic microphones. While the acoustic camera has relatively basic MEMS microphones, it is inexpensive and features a high degree of integration and correspondingly easy operation. Combining the complementary strengths of the two arrays is the key advantage of the present approach over previous approaches as it allows for a quantum leap in accuracy (3.4–4.8 mm, 91% assignment rate), while keeping the complexity of the system manageable. A comparable alternative might be a 16-channel array from high-quality microphones, which would, however, be substantially more expensive (~€40,000) as well as cumbersome to build and refine. A future generation of MEMS microphones might make the use of the high-quality microphones unnecessary and thus further simplify the system setup, allowing for inexpensive, small-form factor deployment (see below).

### Expected impact for future research

Mice and rats are social animals (*Shemesh et al., 2013*; *Lee and Beery, 2019*), and isolated housing (*Kappel et al., 2017*) or testing (*Kondrakiewicz et al., 2019*) can affect subsequent research outcomes. Social isolation also has direct effects on the number and characteristics of USVs, at least in males (*Keesom et al., 2017*; *Portfors, 2007*). *Sangiamo et al., 2020* demonstrated that distinct USV patterns can be linked to specific social actions and the latter that locomotion and USVs influence each other in a context-dependent way. Using HyVL, such analyses could be extended to more close-range behaviors, when a substantial fraction of the vocalizations are emitted (see *Figure 1D*). The development of more unrestricted behavioral paradigms, made viable by increased localization precision, will thus also likely prove valuable to the fields of human language impairment and animal behavior. As an added benefit, better USV localization will also likely increase lab animal well-being via (i) more social contact in specific cases where they spend much time with their conspecifics in the testing environment, or when the home environment is the testing environment (e.g., PhenoTyper; Noldus Information Technologies), and (ii) a reduced need for (non-)invasive markers.

Here, we conducted a limited set of showcase analyses on the spatial characteristics of vocalization behavior. As expected, the system was accurate enough to assign vocalizations during many

snout–snout interactions as well as other, slightly more distant interactions, for example, snout contact with the ano-genital region of the dyadic partner. We found the male mice to vocalize most while making snout contact with the abdomen and ano-genital region of the female wild-type. Females vocalized predominantly during snout–snout contact, with the male's snout in front of the female mouse's snout.

This highlights an example of how localization accuracy can shape our understanding of roles in social interaction between mice: a recent, pivotal study (*Neunuebel et al., 2015*) demonstrated that female mice vocalize during courtship interactions. Research from our group (*Oliveira-Stahl et al., 2023*) concluded further that mice primarily vocalize in snout–snout interactions, incidentally the condition that makes assignment the most difficult. While the present results maintain that female mice vocalize, the fraction appears to be lower than previously thought. We, however, emphasize that this conclusion still requires further study under different social contexts, for example, interaction of more mice as in some of the previous studies (*Warren et al., 2021*; *Sangiamo et al., 2020*).

The compact form factor of the HyVL microphone arrays, in particular the Cam64, enables studies of social interaction in home cages. There, rodents are less stressed and likely to exhibit more natural behavior, in particular if the home cage includes enrichments. The relatively low hardware costs for HyVL allows deployment of multiple systems to cover larger and more natural environments. Research in animal communication with other species could also benefit from use of HyVL, for example, with different insects or other vocal animals, as there is little reason to suspect that the performance of HyVL would not extend to lower frequencies. Flying animals, such as bats or birds, could also be studied; however, the subsequent data analysis would have to be extended by one dimension.

## Current limitations and future improvements of the presented system

The millimeter accuracy by HyVL enables the assignment of USVs even during close interaction, certainly including all snout–ano-genital interactions, and many snout–snout interactions. However, certain snout–snout interactions are still too close to reliably assign co-occurring USVs. While the MPI criterion maintains reliability even then, subsequent analysis will be partially biased due to the exclusion of these USVs during the closest interactions. While a further improvement of accuracy may be possible, close inspection of the sound density maps available via beamforming from the Cam64 recordings suggests that the mouse's snout acts as a distributed source: the sound density is rather evenly distributed on it, without a clear internal peak. During free interaction, we noticed that the sound density was co-elongated with the head direction of the mouse and could thus be used as an additional feature to identify the vocalizer. However, this proved unreliable during close interaction, likely due to absorption and reflection of sounds based on the mice's bodies. More advanced modeling of the local acoustics or deep learning might be able to resolve these issues by analyzing interactions where one mouse is known to be silent, for example, by cutting the laryngeal nerves.

The present strategy for combining the estimates from Cam64/Beamforming and USM4/SLIM was chosen as it optimized the reliably assigned percentage of USVs, while minimizing the residual distance. We also tested alternative approaches, for example, using direct beamforming on the combined data from Cam64 and USM4 (unreliable estimates due to mismatch of number of microphones, not further pursued), maximum likelihood combination of estimates (MAE = 7.1 mm) (*Ernst and Banks, 2002*) and making the selection solely depend on the MPI (MAE = 5.2 mm). While each of these approaches has certain, theoretically attractive features, the results were worse in each case, likely due to particular idiosyncrasies of the MPI computation, the different microphone characteristics, and the estimation of single-estimate uncertainty.

A small set of vocalizations was not assigned solely due to the overall proximity threshold of 50 mm (see 'Materials and methods,' 2.9%). We have previously shown that very quiet or very short USVs are, unsurprisingly, harder to detect and localize (*Oliveira-Stahl et al., 2023*). In addition, spectrally narrow and acoustically occluded USVs are likely hard to localize: USVs that are spectrally very narrow – that is, close to a pure tone – will have phase ambiguity, which will make it hard to assign a single location. USVs that are acoustically occluded – for example, an animal vocalizing away from a microphone, or a mouse body in the path of the sound – will have a reduced signal-to-noise ratio (SNR) on one or more microphones. In our experience, the latter two affect the Mic4 data more than Cam64 due to their different placement relative to the platform.

A very small percentage of vocalizations (<0.1%) contained multiple, differently shaped vocalization traces that, when reanalyzed in shortened time-frequency bins with beamforming, could be assigned to two different males. Such overlapping vocalizations did not form a harmonic stack. Overall, overlaps were surprisingly rare and only occurred when our USV detection algorithm produced a longer interval, affecting the cumulative heatmap because beamforming is separately performed from the onset to the end of each vocalization. Although the identity of the assigned vocalizer could shift in these very rare cases depending on which time bin was reanalyzed, the system's localization performance remained in principle unaffected: as mentioned above, shorter time bins on nonoverlapping parts correctly show the origin of the vocalizations in this case, and we think that improved USV detection/separation based on the harmonic structure will partially address this issue. During the beamforming, each vocalization can then be separately localized by restricting the beamforming to the corresponding time *and* frequency range. Further, the beamforming analysis could be refined so that multiple salient peaks can be detected in the soundfield estimate, for example, a sequence of soundfield estimates would be computed on shorter segments of data and later fused again. As this uses less data per single estimate, it also increases the possibility of false positives, which in the current situation with very few overlaps in time would likely reduce the overall accuracy of the system. Lastly, for the present data, if a time window was analyzed such that the intensity map of the sound field contains multiple hotspots of an approximately equal magnitude, the USV would likely remain unassigned because the within-soundfield uncertainty would be higher than for a single peak, and this would reduce the MPI. However, given the rarity of these cases in our dataset, we do not think that their exclusion would change the results appreciably.

Lastly, for the purpose of online feedback during experiments and to reduce data warehousing, it would be advantageous to perform the localization of USVs in real time. This would be enabled by streaming the data to a GPU, performing localization immediately and keeping only a single channel, beamformed estimate of each USV. Ideally, the same device could run visual tracking simultaneously, which would remove all temporal limitations on the recordings in terms of data size and enable continuous audiovisual tracking.

## Conclusion and outlook

HyVL delivers breakthrough accuracy and assignment rates, likely approaching the physical limits of assignment. The low system costs (<€10k) in relation to its performance make HyVL an excellent choice for labs studying rodent social interaction. Many recent questions regarding the sequencing of vocalizations during social interactions become addressable with HyVL without intrusive interventions. Its use can both refine the precision and reliability of the analysis, while reducing the number of animals required to complete the research due to a larger fraction of assigned USVs per animal.

# Materials and methods

**Key resources table**

| Reagent type (species) or resource | Designation | Source or reference | Identifiers | Additional information |
|---|---|---|---|---|
| Transfected construct (*Mus musculus*) | Foxp2$^{flox/flox}$;Pcp2$^{Cre}$ | Bred locally at animal facility | | |

All experimental procedures were approved by the animal welfare body of the Radboud University under the protocol DEC-2017-0041-002 and conducted according to the Guidelines of the National Institutes of Health.

## Animals

In our experiment, four female C57Bl/6J-WT, six male C57Bl/6J-WT, and eight male C57Bl/6J-Foxp2$^{flox/flox}$;Pcp2$^{Cre}$ mice (bred locally at the animal facility) were studied. For subsequent analyses, WT and KO mice were combined (see beginning of 'Results' for reasoning). The mice were 8 weeks old at the start of the experiments. After 1 wk of acclimation in the animal facility, the experiments were started. Mice of the same sex were housed socially (2–5 mice per cage) on a 12 hr light/dark cycle with ad libitum access to food and water in individually ventilated, conventional EU type II mouse cages at 20°C with paper strip bedding and a plastic shelter for basic enrichment. Upon completion of the experiments, the animals were anesthetized using isoflurane and sacrificed using $CO_2$.

The current experiment was performed as an add-on to an existing set of experiments, whose focus included a region-specific knockout of *Foxp2* in the cerebellar Purkinje cells of the male mice, denoted as Foxp2$^{flox/flox}$;Pcp2$^{Cre}$. Neither previous work nor our own work has detected any differences in USV production between WT and KO animals (*Urbanus et al., 2020*), so – given the mostly methodological focus of the present work – we considered it acceptable to pool them in the current analysis, reducing the number of animals needed, thus treating all males as WT C57Bl/6J, the genotype of the female mice.

## Recording setup

The behavioral setup consisted of an elevated interaction platform in the middle of an anechoic booth together with four circumjacent ultrasonic microphones as well as an overhanging 64-channel microphone array and high-speed video camera (see *Figure 1A*).

The booth had internal dimensions of 70 × 130 × 120 cm (L × W × H). The walls and floor were covered with acoustic foam on the inside (thickness: 5 cm, black surface Basotect Plan50, BASF). The acoustic foam shields against external noises above ~1 kHz with a sound absorption coefficient >0.95 (N.B., defined as the ratio between absorbed and incident sound intensity), which corresponds to >26 dB of shielding apart from the shielding provided by the booth itself. In addition, the foam strongly attenuates internal reflections of high-frequency sounds like USVs. Illumination was provided via three dimmable LED strips mounted to the ceiling, providing light from multiple angles to minimize shadows.

The support structure for the interaction platform and all recording devices was a common frame constructed from slotted aluminum (30 × 30 mm) mounted to the floor of the anechoic booth, guaranteeing precise relative positioning throughout the entire experiment. The interaction platform itself was a 40 × 30 cm rectangle of laminated, white acoustic foam (thickness 5 cm; Basotect Plan50) chosen to maximize the visual contrast with the mice and simplify the cleaning of excreta. The interaction platform had no walls to avoid acoustic reflections and was located centrally in the booth. Its surface was elevated 25 cm above the floor (i.e., 20 cm above the foam on the booth floor), which was generally sufficient in preventing animals from leaving the platform. If a mouse left the platform, data was excluded from further analysis (<5% of frames).

Sounds inside the booth were recorded with two sets of microphones: (i) four high-quality microphones (USM4) and (ii) a 64-channel microphone array (Cam64), both recording at a sampling rate of 250 kHz at 16 bits. (i) The four high-quality microphones (CM16/CMPA48AAF-5V, AviSoft, Berlin) were placed in a rectangle that contained the platform (see *Figure 1A*) at a height exceeding the platform by 12.1 cm to minimize the amount of sound blocked by the mice during interaction. The position of a microphone was defined as the center of the recording membrane. Considering the directional receptivity of the microphones (~25 dB attenuation at 45°), the microphones were placed a short distance away from the corners of the platform to maximize sound capture (5 cm in the long direction and 6 cm in the short direction of the platform). The rotation of each microphone was chosen to be such that it aimed at the platform center. The microphones produce a flat (±5 dB) frequency response within 7–150 kHz that was low-pass filtered at 120 kHz to prevent aliasing (using the analog, 16th-order filter, which is part of the microphone amplifier). Recorded data was digitized using a data acquisition card (PCIe-6351, National Instruments). (ii) In addition, a 64-channel microphone array (Cam64 custom ultrasonic version, Sorama B.V.) was mounted above the platform with a relative height of 46.5 cm measured to the bottom of the Cam64 and a relative lateral shift of 6.52 cm to the right of the platform midpoint. The Cam64 utilizes 64 MEMS microphones (Knowles, Digital Zero-Height SiSonic, SPH0641LU4H-1) for acoustic data collection that are positioned in a Fermat's spiral over a circle with an ~16 cm diameter. Raw microphone data was streamed to an m.2 SSD for later analysis. Synchronization between the samples acquired by the Cam64 and the ultrasonic microphones was performed by presenting two brief acoustic clicks (realized by stepping a digital output from 0 to 5 V) close to one of the microphones on the Cam64 at the start and end of each trial using a headphone driver (IE 800, Sennheiser). The recorded pulses were automatically retrieved and used to temporally align the recording sources.

A high-speed camera (PointGrey Flea3 FL3-U3-13Y3M-C, Monochrome, USB3.0) was mounted above the platform with a relative height of 46.5 cm measured to the bottom of the front end of the lens (6 mm, Thorlabs, part number: MVL6WA) and a relative lateral shift of 4.48 cm to the left of the

platform midpoint. Video was recorded with a field of view of 52.2 × 41.7 cm at ~55.6 fps (18 ms interframe interval) and digitized at 640 × 512 pixels (producing an effective resolution of ~0.815 mm/pixel). The shutter time was set to 10 ms to guarantee good exposure while keeping the illumination rather dim. The frame triggers from the camera were recorded on an analog channel in the PCIe-6531 card for subsequent temporal alignment with the acoustic data.

## Experimental procedures

The experiment had three conditions: dyadic (with two mice), triadic (with three mice), and monadic (single male mouse, one type of ground truth data). For each of the male animals (n = 14), we conducted one trial with each female (n = 4) in dyadic and triadic conditions, that is, 112 trials in total, in pseudo-random order. The third animal in triadic conditions was chosen pseudo-randomly. Afterwards, to maximize the number of USVs for evaluation of the localization system, another 108 trials were run with the best male vocalizers in both dyadic and triadic conditions, leading to a total of 220 trials. In 85/220 trials, USVs were emitted by the mice (57 dyadic, 28 triadic), prompting the experimenter to initiate a Cam64 recording (see below). Two dyadic trials were excluded from further analysis due to repeated but required experimenter interference during the recordings leaving 55 dyadic trials. The USVs from the remaining 83 trials formed the basis for the evaluation of the tracking accuracy of HyVL, while we used the 112 balanced-design dyadic and triadic recordings (with and without USVs) in the analysis of differences in dyadic/triadic interactions (*Figure 6*). Lastly, eight trials were recorded with just a single male mouse on the platform.

Each trial consisted of 8 min of free interaction between at least one female and at least one male mouse on the platform. Females were always placed on the platform first, and males were added shortly thereafter. In the monadic case, fresh female urine was placed on the platform instead of a female mouse to prompt the male mouse to vocalize. The high-speed camera and four high-quality microphones started recording after all mice had been placed on the platform and continued for 8 min. Data points where one mouse had left the platform or the hand of the experimenter were visible 10 s before or after (e.g., to pick up a mouse) were discarded (<5% of frames). Due to the rate of data generation of the Cam64 recordings (32 MB/s), their duration and timing were optimized manually. The experimenter had access to the live spectrogram from the USM4 microphones, and upon the start of USVs, triggered a new Cam64 recording (of fixed 2 min duration). If additional USVs occurred after that point, the experimenter could trigger additional recordings.

## Data analysis

The analysis of the raw data involved multiple stages (see *Figure 2*): from the audio data, the presence and origin of USVs were estimated automatically. From the video data, mice were carefully tracked by hand at the temporal midpoint of each USV as near-optimal estimates for their acoustically localized origin. To estimate what proportion of our precision would be lost when using a faster and more scalable visual tracking method, we also tracked the mice automatically during dyadic trials. The estimated locations of the mice and USVs were then used to attribute the USVs to their emitter. All these steps are described in detail below.

### Audio preprocessing

Prior to further analysis, acoustic recordings were filtered at different frequencies. USM4 data was band-pass filtered between 30 and 110 kHz before further analysis using an inverse impulse response filter or order 20 in MATLAB (function: designfilt, type: bandpassiir). Cam64 data was band-pass filtered with a frequency range adapted to the frequency content of each USV. Specifically, first the frequency range of the USV was estimated as the 10th–90th percentile of the set of most intense frequencies at each time point. Next, this range was broadened by 5 kHz at both ends, and then limited at the top end to 95 kHz. If this range exceeded 50 kHz, the lower end was set to 45 kHz. This ensured that beamforming was conducted over the relevant frequencies for each USV and avoided the high-frequency regions where the Cam64 microphones are dominated by noise (see *Figure 1C*, *Figure 1—figure supplement 1*).

## Video preprocessing

The high-speed camera lens failed to produce perfect rectilinear mapping and was placed off-center with respect to the interaction platform, thereby producing a nonlinear radial-tangential visual distortion. We corrected for the radial distortion with

$$x_{rd} = x_{\frac{1}{2}} + \frac{atan\left(r_d/\lambda\right)}{r_d/\lambda} * \left(x_{ru} - x_{\frac{1}{2}}\right) * Z_x$$

$$y_{rd} = y_{\frac{1}{2}} + \frac{atan\left(r_d/\lambda\right)}{r_d/\lambda} * \left(y_{ru} - y_{\frac{1}{2}}\right) * Z_y$$

where $\left[x_{rd}, y_{rd}\right]$ represent the radially distorted image coordinates, $\left[x_{\frac{1}{2}}, y_{\frac{1}{2}}\right]$ the coordinates of the image center, $r_d$ the Euclidean distance to the radial distortion center, $\lambda$ the distortion strength, $\left[x_{ru}, y_{ru}\right]$ the radially undistorted coordinates, and $Z_x, Z_y$ axis-specific zoom factors. The tangential distortion, on the other hand, we corrected with

$$x_{td} = x_{tu} - \frac{(x_{tu} - a_x)}{|x - a_x|} * \frac{\kappa_x}{p_y} * (y_{tu} - \Delta p_y) * Z_x$$

$$y_{td} = y_{tu} - \frac{(y_{tu} - a_y)}{|y - a_y|} * \frac{\kappa_y}{p_x} * (x_{tu} - \Delta p_x) * Z_y$$

where $\left[x_{td}, y_{td}\right]$ represent the tangentially distorted image coordinates, $\left[x_{tu}, y_{tu}\right]$ the tangentially undistorted coordinates, $\left[a_x, a_y\right]$ the coordinates of the tangential distortion center, $[x, y]$ the size of the image, $\left[\kappa_x, \kappa_y\right]$ the tangential distortion strengths, $\left[p_x, p_y\right]$ the size of the interaction platform in the undistorted image, and $\left[\Delta p_x, \Delta p_y\right]$ the offset of the platform with respect to the top-left corner of the undistorted image.

## Detection of ultrasonic vocalizations

USVs were detected automatically using a set of custom algorithms described elsewhere (*Ivanenko et al., 2020*). Detection was only performed on the USM4 data as their sensitivity and frequency range were generally better than for the Cam64 (see *Figure 1C*, *Figure 1—figure supplement 1*). A vocalization only had to be detected on one of the four high-quality microphones to be included into the set. In total, we collected 13,406 USVs, out of which 8424 occurred when the Cam64 recordings were active.

## Automatic visual animal tracking

To assess whether we could reliably assign USVs to their emitter in a fast and scalable way, we automatically tracked multiple body parts of interacting mice in all frames — most importantly the snout and head center — for all dyadic trials (using *DeepLabCut* [*Brudzynski, 2021*]; see *Figure 2*) and a subset of triadic trials (using SLEAP [*Pereira et al., 2022*]; see *Figure 6*). With this approach, tracking is not temporally restricted to the midpoint of USV production, but can be performed for every frame of the entire recording. This data can be used to establish spatial densities of interaction against which, for example, the spatial density of vocalizations can be compared (*Oliveira-Stahl et al., 2023*).

For the dyadic recordings, mice were tracked offline using a combination of *DeepLabCut (DLC)* (*Mathis et al., 2018*) and extensive pos-processing to maintain animal identity over the entire recording. While the tracking results from DLC were generally quite accurate, we refrained from using them directly because of inaccuracies and identity switches that occurred on many hundreds of occasions in every recording. Instead we adopted a strategy where DLC generated an overcomplete set of candidate locations followed by custom synthesis and tracing of these alternatives in space and time (see *Figure 3—figure supplement 1*). In short, improved marker locations were generated from marker estimate clouds produced by DLC. Next, these marker positions were assembled into short spatiotemporal threads with the same, unknown identity based on a combination of spatial and temporal analysis. Finally, the thread ends were connected based on quadratic spatial trajectory estimates for each marker, yielding the complete track for both mice. This strategy resulted in

**Video 1.** Example of Hybrid Vocalization Localizer (HyVL) tracking and sound localization. Marker color represents animal sex (light blue: male; light red: female). Marker shape represents body part (circle: body center; cross: snout or tail; downward triangle: left ear; upward triangle: head center; diamond: right ear). Cam64 ultrasonic vocalization (USV) localizations (yellow) are overlaid on the beamforming densities (red) that are often very narrow and therefore hard to see underneath the localization marker (yellow dot). SLIM USV localizations are shown as well (orange '+'), typically further away from the snout in comparison to Cam64-based localization markers.

https://elifesciences.org/articles/86126/figures#video1

reliable, high-quality tracking for all recordings, with a greatly reduced number of manual corrections needed overall (~10 per trial on average). All resulting tracks were visually verified (for a representative example, see *Video 1*).

For tracking the triadic interactions with two males, we used the SLEAP (*Pereira et al., 2022*) tracking system (version 1.3.0). To obtain the frame-by-frame pose estimations, we utilized the SLEAP graphical user interface to train a bottom-up U-net model, which is used to identify the body parts first and then attribute them to separate instances. Initially, we trained the model on the manually annotated frames from the dyadic tracking process. Subsequently, we annotated ~200 additional frames, all in triadic conditions in which the model exhibited poor performance. The extended dataset was then used to retrain the model. To establish the basis for triadic tracking, we employed SLEAP's tracker to group the predicted instances across frames. The tracker compared instances across the full six-node skeleton and aimed to maximize the overall similarity across the three track assignments using the Hungarian algorithm. To identify candidate instances for comparison, it employed optical flow based on the previous five frames and selected instances based on the 0.95 quantile of similarity scores. We also applied SLEAP's post-tracking data cleaning techniques to connect any breaks in single tracks. Subsequently, we examined all 14 recordings frame by frame to rectify any identity switches and eliminate inaccurate predictions. For instance, we addressed cases where two instances were detected on a single mouse or when one instance appeared to cover two mice. To further refine the results, we interpolated outlying instances based on velocity jumps.

We compared the accuracy of localization on the basis of manual tracking with that of automatic tracking (N = 5046 USVs, see *Figure 3—figure supplement 3*). Directly comparing the snout positions between the methods shows a median difference of 3.76 mm. The resulting error for localizing USVs was still superior to other systems, but significantly increased by ~0.9 mm (MAE = 5.71 mm) relative to manual tracking. Both manual and automatic tracking appear to have particular patterns of residual errors, indicated by the fact that the error between the tracking methods is much larger than their difference in USV localization error. The percentage of reliably assignable USVs interestingly increased to 93.6% (HyVL) compared to 92% with manual tracking for the dyadic recordings only. We optimized the mouth location on the snout-to-head-center line, finding an optimal distance of 15% of the snout to head center distance *to the front* of the animal. This indicated that the automatic tracking tended to place the snout tracking point a bit further into the snout than manual tracking, which might also explain the increase in assignment, due to a slight – but erroneous – increase in the separation between the snouts. While these results suggest that manual tracking is still advantageous, it highlights that completely automatic analysis of dyadic and possibly n-adic social interaction experiments is feasible at slightly reduced accuracy.

## Manual visual animal tracking

To test the maximum precision of HyVL, we manually tracked the spatial locations of all mice during all USVs from the video data to assess the precision of the automatic visual and acoustic tracking. During manual tracking, the observer was presented with a combined display of the vocalization spectrogram and the concurrent video image at the temporal midpoint of each USV (*MultiViewer*, custom-written, MATLAB-based visualization tool). The display included a zoom function for optimal accuracy as tracking was click-based. Users could also freely scroll in time to ensure consistent animal identities. Only the snout and head center (i.e., midpoint between the ears) needed to be annotated

because these points define a vector representing the head location and direction, which was all that was required in subsequent behavioral analyses.

## Localization of ultrasonic vocalizations

USVs were spatially localized using a hybrid approach that integrates SLIM (*Oliveira-Stahl et al., 2023*) (based on four high-quality microphones) and beamforming (based on the 64-channel microphone array), drawing on the complementary strengths of the two microphone arrays (see *Figure 1—figure supplement 1*). For example, the Cam64 array provided excellent localization for USVs with energy below ~90 kHz due to the increasing noise floor of the MEMS microphones with sound frequency. Conversely, the four high-quality ultrasonic microphones (USM4) have a rather flat noise level as a function of frequency. On the other hand, USM4 will occasionally have glitches in one of the microphones, which can be compensated for in Cam64-based estimates through the number of microphones. As a consequence, the errors of the two methods show an L-shape (see *Figure 3A*), which highlights the synergy of a hybrid approach.

Acoustic localization using the Cam64 recordings was performed on the basis of delay-and-sum beamforming (*Van Veen and Buckley, 1988*). In beamforming, signals from all microphones are combined to estimate a spatial density that correlates with the probability of a given location being the origin of the sound. Specifically, we computed beamforming estimates for a surface situated 1 cm above and co-centered with the interaction platform, extending to 5 cm beyond all edges of the platform (i.e., 50 × 40 cm in total) at a final resolution of 1 mm in both dimensions. We refer to this density of sound origin as $D_{SO}(x, y)$ where $x$ and $y$ denote spatial coordinates. To prevent noises unrelated to a specific USV from contaminating the location estimate, we limited beamforming to a particular frequency range estimated from the simultaneous data of the USM4 array that enveloped the USV. Spatial density was defined as

$$D_{SO}(x, y) = \sum_{f=F_{min}}^{F_{max}} D_{SO}(x, y, f) = \sum_{f=F_{min}}^{F_{max}} \sum_{m=1}^{64} e^{i2\pi f d(m, x, y, z)}$$

where $d(m, x, y, z)$ denotes the difference in arrival time at each microphone $m$ for sounds emitted from a location with coordinates $(x, y, z)$, where $z$ is omitted in $D_{SO}(x, y)$ as it is a fixed distance to the plane of the microphone array. Beamforming was performed in the computational cloud backend provided by the Cam64 manufacturer, the so-called Sorama Portal (https://www.sorama.eu/sorama-portal).

The final beamforming estimate was calculated sequentially in two steps: first, a coarse estimate with 1 cm resolution was generated over the entire beamforming surface. Second, a fine-grained estimate with 1 mm resolution was generated over a 30 × 30 mm window centered on the peak location of the coarse estimate (see *Figure 2* for an example). This two-step approach was chosen to optimize performance, as an estimate with 1 mm resolution over the entire beamforming surface would be computationally expensive while failing to produce a better result. For USVs of sufficient quality (i.e., containing frequency content below ~90 kHz while being sufficiently intense and long), both the coarse and fine estimates of $D_{SO}(x, y)$ contained a peak whose height was typically very large compared to the surrounding values at distances greater than a few centimeters. The peak location of the fine-grained estimate was used as the final estimate of the USV's origin. To assess the quality of this location estimate, we computed a SNR per USV as follows:

$$SNR_{Cam64}(v) = \frac{max(D_{SO}(x, y))}{std(D_{SO}(x, y))}$$

where $D_{SO}(x, y)$ is assumed to be calculated for the USV $v$. The inverse, $1/SNR_{Cam64}$ was used as a proxy for the uncertainty of localization for a given USV.

Localization from the USM4 recordings was performed using the SLIM method (*Oliveira-Stahl et al., 2023*). Briefly, SLIM analytically estimates submanifolds (in 2D: surfaces) of a sound's spatial origin for each pair of microphones and combines these into a single estimate by intersecting the manifolds (in 2D: lines). The intersection has an associated uncertainty that scales with the uncertainty of the localization estimate for a given USV, specifically the uncertainty was defined as the standard deviation of all locations that were >90% times the maximum of the intersection density of all origin curves.

Lastly, for each USV where both Cam64 and SLIM location estimates $\dot{X}_{Cam64}$ and $\dot{X}_{SLIM}$ were available, a single estimate $\dot{X}_{HyVL}$ was computed based on the two estimates, spatial uncertainties and their spatial relation to the mice at the current time (see below).

## USV assignment

The final hybrid location estimate and assignment to a mouse was performed while taking into account the probability of making a false assignment as proposed before (*Neunuebel et al., 2015*), through the calculation of the mouse probability index *MPI*. While the *MPI* was previously only used to exclude uncertain assignments (e.g., if two mice are nearly equidistant to the estimated sound location), we also adapted it here to select and combine the location estimates. The $MPI_k$ for each mouse $k$ was computed as,

$$MPI_k = \frac{P_k}{\sum_{m=1}^{n} P_m}$$

Here, $P_k$ is the probability that the USV in question originated from mouse $k$ computed as $P_k = N\left(\dot{X}_{Method} - X_{mouth,k}, \sigma_{Method}^2\right)$ , where $\dot{X}_{Method}$ is an estimate of the acoustic origin, $X_{mouth,k}$ the position of the mouth of mouse $k$, and $\sigma_{Method}^2$ the uncertainty of the estimate, with $_{Method}$ and $\sigma_{Method}^2$ specific to the Method used. $X_{mouth,k}$ was assumed to lie on a line connecting the snout and head-center. For manually tracked recordings, the optimal location on this line was close to the snout (~2% toward the head, where % is relative to the snout-to-head-center tracked distance), while in the automatic tracking it was ahead of the snout tracking point (~15% away from the head). $\sigma_{Method}^2$ was computed for each USV as the method's intrinsic per-USV uncertainty estimate. As these uncertainty estimates only correlate with the absolute uncertainty (i.e., in millimeters), we scaled them such that their average across all USVs matched the residual error of each method in the Far-condition (all animals >100 mm apart, see *Figure 3C* and *Oliveira-Stahl et al., 2023*). In this way, the $MPI_k$ for individual USVs took into account the uncertainty of each method: if the uncertainty of one method was higher, probabilities across mice would become more similar and the $MPI_k$ would reduce.

For a given USV, we computed the $MPI_k$ for all mice for both methods. The mouse with the largest $MPI_k$ per method, which coincides with the mouse at the smallest distance to the estimate, was denoted as $MPI_{Cam64}$ and $MPI_{SLIM}$ , respectively. If only one of the two exceeded 0.95, this method's estimate was selected. If both exceeded 0.95, then the estimate with the smaller distance to the mouse with the highest $MPI_k$ was chosen. This combination ensured that only reliable assignments were performed, while minimizing the residual error. Similar to *Neunuebel et al., 2015*, we also excluded estimates that were too far away from any mouse (50 mm). This distance threshold mainly serves to compensate for a deficiency of the *MPI*: if all mice are far from the estimate, all $P_k$ are extremely small; however, the $MPI_k$ will often exceed 0.95. The distance threshold corresponds to setting the individual $P_k = 0$ in the $MPI_k$ , thus excluding candidate mice that are highly unlikely to be the source of the USV. USVs that had no $MPI_k > 0.95$ for either method were excluded from further analysis. The fraction of included USVs is referred to as *selected* in the plots. Maximizing this fraction is essential to perform a complete analysis of vocal communication.

We compared the above-described combination strategy to a large number of alternative strategies, including maximum likelihood combination of estimators (*Ernst and Banks, 2002*), or selecting directly based on the largest $MPI_k$ or largest $P_k$ . While all these approaches led to broadly similar results, the described approach achieved the most robust and reliable results (see 'Discussion' for additional details).

## Audiovisual alignment

For both microphone sets, precise measurements of their location in relation to the camera's location were used to position acoustic estimates in the coordinate system of the images provided by the camera. In the final analysis, we noticed for each microphone set small, systematic (0.5–2 mm) shifts in both X and Y. We interpreted these as very small measurement errors in the relative positions of the camera or microphone arrays and corrected these post hoc in the setup definition, followed by rerunning all subsequent analysis steps. This reduced all systematic shifts to near 0.

## Spatial vocalization analysis

To gain insight into the spatial positioning of the interacting mice, we represented the relative animal positions in a polar reference frame centered on the snout of the emitter. In this format, the radial distance corresponded to the snout–snout distance and the radial angle described the relative angle between the gaze direction of the emitter and the snout position of the recipient (i.e., with the line from the head center to the snout of the emitter pointing towards 0°; see also *Figure 4E*).

The position density of the recipient mouse was collected in cumulative fashion, with the polar coordinate system translated appropriately for each USV based on its temporal midpoint. We assumed that the mice had no preference for relative vocalizations to either side of their snout, so all relative spatial positions were agglomerated in the right hemispace for further analysis. All data points were then binned using a polar, raw-count histogram with bins of 10° and 1 cm.

## Statistical analysis

To avoid distributional assumptions, all statistical tests were nonparametric, that is, Wilcoxon rank-sum test for two-group comparisons and Kruskal–Wallis for single-factor ANOVA. Correlations were computed as Spearman's rank-based correlation coefficients. Error bars represent standard errors of the mean (SEM) unless stated otherwise. All statistical analyses were performed in MATLAB v.2018b (The MathWorks, Natick) using functions from the Statistics Toolbox.

## Acknowledgements

We thank Lucas Noldus for suggesting the use of the Sorama Cam64 and Maurice Camp and Toros Senan for technical support relating to the operation and data handling of the Cam64 and the Sorama Portal. We would like to thank Amber van der Stam, Dionne Lenferink, and Soha Farboud for assisting with the animal handling.

## Additional information

### Funding

| Funder | Grant reference number | Author |
| --- | --- | --- |
| Noldus IT | DCN Internal Grant | Bernhard Englitz |
| NWO VIDI grant | 016.VIDI.189.052 | Bernhard Englitz |
| ZonMw | Technology Hotel Grant 40-43500-98-4141 | Bernhard Englitz |

The funders had no role in study design, data collection and interpretation, or the decision to submit the work for publication.

### Author contributions

Max L Sterling, Software, Formal analysis, Validation, Investigation, Visualization, Methodology, Writing - original draft, Writing - review and editing; Ruben Teunisse, Formal analysis, Writing - review and editing; Bernhard Englitz, Conceptualization, Data curation, Software, Formal analysis, Supervision, Funding acquisition, Investigation, Visualization, Methodology, Writing - original draft, Project administration, Writing - review and editing

### Author ORCIDs

Max L Sterling http://orcid.org/0000-0002-2114-2265
Bernhard Englitz http://orcid.org/0000-0001-9106-0356

### Ethics

All of the animals and experimental procedures were conducted according to the guidelines of the Animal Welfare Body of the Central Animal Facility at the Radboud University. The protocol was approved by the Dutch National Committee CCD (Permit Number: 2017-0041).

Decision letter and Author response
Decision letter https://doi.org/10.7554/eLife.86126.sa1
Author response https://doi.org/10.7554/eLife.86126.sa2

## Additional files

### Supplementary files
• MDAR checklist

### Data availability
All code necessary to implement the HyVL system has been deposited at https://github.com/benglitz/HyVL (copy archived at *Englitz, 2023*) and https://doi.org/10.34973/7kgc-ta72. All data has been made available at https://doi.org/10.34973/7kgc-ta72.

The following dataset was generated:

| Author(s) | Year | Dataset title | Dataset URL | Database and Identifier |
|---|---|---|---|---|
| Sterling M, Englitz B, Teunisse R | 2023 | Ultrasonic vocal interaction resolved with millimeter precision using hybrid beamforming | https://doi.org/10.34973/7kgc-ta72 | Donders Repository, 10.34973/7kgc-ta72 |

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
