## [Editor Report]

This study introduces a novel and important hybrid strategy for recording ultrasonic vocalizations by combining data from several high quality microphones with data from a dense array of less sensitive microphones. This method enables recordings to be made from pairs and trios of freely interacting mice and accurate localization of their point of origin to convincingly determine the identity of the caller for each vocalization. This technology opens the door to new experiments incorporating analysis of vocal communication into behavioral paradigms.

---

## [Decision Letter]

**Decision letter after peer review:**

Thank you for submitting your article "Rodent ultrasonic vocal interaction resolved with millimeter precision using hybrid beamforming" for consideration by *eLife*. Your article has been reviewed by 3 peer reviewers, including Brice Bathellier as Reviewing Editor and Reviewer #1, and the evaluation has been overseen by Andrew King as the Senior Editor. The following individuals involved in the review of your submission have agreed to reveal their identity: Stephen D Shea (Reviewer #2); Elodie Ey (Reviewer #3).

Essential Revisions (for the authors):

1) Clarify the relationship of the present manuscript to https://www.biorxiv.org/content/10.1101/2021.10.22.464496v1

2) Provide ground truth accuracy measurements based on known sources (loudspeakers).

3) Provide more explanations about the impact of obstacles (e.g. other animals).

4) Clarify the impact of pre-experiment social isolation on vocal behavior.

5) Release the code online.

6) Detail male-female interactions as requested by reviewer 3 (vocalization during anogenital sniffing?). Clarify tryadic interactions.

7) Improve statistics as suggested by reviewer 3 and address all small clarification comments.

*Reviewer #1 (Recommendations for the authors):*

– The calibration of the mouse probability index against ground truth data would be a useful addition to the study, to evaluate if the estimator has biases in particular situations in which the method e.g. fails more often.

– The authors find that females vocalize at a level close to the emitter identification accuracy. The authors discuss that but could be more clear whether females vocalize or not: comparing male-female USVs, showing female USVs in isolation or far from the male even if this is extremely rare.

*Reviewer #3 (Recommendations for the authors):*

The manuscript is well written. Here are a few comments that could help to clarify some points:

l. 92-93 + l. 439-442: The overestimation of female vocalisations during male-female encounters in previous studies might also be related to the fact that the animals used were socially isolated for at least two weeks before the experiments (for instance in Neunuebel et al. 2015, *eLife*; Sangiamo et al. 2020 Nat Neurosci). This type of isolation is drastic for females and leads to an increase in social motivation, and therefore ultrasonic vocalisations emission even when interacting with females (see Ey et al. 2018 Frontiers Mol Neurosci).

The proportions cited in the main text (l. 195-207) and in the legend of Figure 3 are not corresponding to the ones depicted in the graphs of Figure 3.

l. 195-207: Is this analysis only for dyadic interactions? If yes, this should be specified in the figure legend. Maybe the same analysis conducted with triadic interactions could be conducted in a supplementary figure?

l. 216: Could the authors explain why a small proportion of USVs cannot be assigned? Were the USVs too soft? Did they include specific acoustic characteristics that render them difficult to localise?

l. 231: Could the system be applied to other species? Which ones? With what types of adaptations?

l. 259: References should be added after the citation of the proportions.

l. 262: There are two times the word "overall" in the sentence.

l. 262-266 and Figure 4A and B: Are the analysis conducted for dyadic and triadic interactions together?

l. 314-315: As the tracking also depicts the tail basis, would it be possible to add a plot of the distance between the emitter's snout and the receiver's tail basis, to confirm the result that males vocalise mostly during anogenital sniffing of the female? This could be conducted at least on a subset of data.

l. 331: The authors mention a Bonferroni correction for multiple testing. Could they precise how many tests they conducted, over which data…

Figures 4 and 5: The legend should provide more information about the sample sizes of each condition and what represents each point (an individual?). As stated before, the individual should be taken as a factor in the statistical analyses.

Figure 5: How were females separated into dominant and subordinate? Just based on the call rate like males?

l. 331-334: To better justify the separation between dominant and subordinate that is done here based on the call rate, could the authors confirm this status with behavioural markers such as approaches/escapes, types of approaches, proximity to other individuals, and other measures?

l. 370-371: In Figure 5D, the distance to the closest female is shorter for the subordinate compared to the dominant in triadic encounters. This does not seem to fit with the statement that the dominant gets closer to the receiver when vocalising.

Figure 5C and 5D: As the females are emitting few vocalisations, are the analyses of mean energy and distance to the closest individual meaningful for females? Could the authors provide the sample size (number of vocalisations, number of individuals)?

l. 433-434: Does that mean that females turn around on themselves before vocalising? How have the authors estimated the approach of the male? Is it based on the direction of the animals? If yes, maybe the authors could reformulate the sentence by saying that females vocalise during snout-snout contact while being oriented in the same direction as a receiver or something equivalent.

l. 463: Given USVs production mechanisms that are discussed (i.e., Boulanger-Bertolus and Mouly (2021) Brain Sci), cutting vocal folds will likely not be sufficient, but maybe cutting laryngeal nerves (Pomerantz et al. 1983 Physiology and Behavior).

l. 563: What is the light intensity?

l. 580: "fixed 2 min duration": This is also an issue to be solved for continuous monitoring.

---

## [Author Response]

Essential Revisions (for the authors):1) Clarify the relationship of the present manuscript to https://www.biorxiv.org/content/10.1101/2021.10.22.464496v1

See response to point 1 by Reviewer 2 in Public Review.

2) Provide ground truth accuracy measurements based on known sources (loudspeakers).

See point 2 by Reviewer 2 in Public Review.

3) Provide more explanations about the impact of obstacles (e.g. other animals).

See response to point 4 by Reviewer 2 in Public Review.

4) Clarify the impact of pre-experiment social isolation on vocal behavior.

See Response to first major point in recommendations to Reviewers by Reviewer 3.

5) Release the code online.

See response to point 6 by Reviewer 2 in Public Review.

6) Detail male-female interactions as requested by reviewer 3 (vocalization during anogenital sniffing?). Clarify tryadic interactions.

See responses to comment relating to lines 314-315 by Reviewer 3.

Reviewer #1 (Recommendations for the authors):– The calibration of the mouse probability index against ground truth data would be a useful addition to the study, to evaluate if the estimator has biases in particular situations in which the method e.g. fails more often.

This is an interesting idea, and we agree that it would be useful, however, we have come to conclude that only an experiment with two mice in which one mouse is devocalized can provide the corresponding data to address this question (for which we do not have an ethical permit). In our opinion, basing such an MPI investigation on a speaker that produces vocalizations would not likely lead to a trustable outcome, due to the manifold differences between speaker generated sounds and mouse generated ones (e.g. , among others). Likely, results from a speaker would likely lead to an unreasonably low variability and thus unreasonably low rate of failure/misattribution, which does not translate to actual mouse vocalizations (see the new Supplementary Figure 4, where we demonstrate that the accuracy for a speaker is likely much higher than for a mouse).

– The authors find that females vocalize at a level close to the emitter identification accuracy. The authors discuss that but could be more clear whether females vocalize or not: comparing male-female USVs, showing female USVs in isolation or far from the male even if this is extremely rare.

This is an excellent suggestion, and we had partially demonstrated this in Figure 4C, showing that the fraction of female vocalizations further decreases substantially and significantly for instances where the snouts are far apart (relative to the localization accuracy of HvVL, i.e. >50mm).

However, to give more insight into the instances where female vocalization appears most accurate at far distances, we filtered all female vocalizations and ranked them on the basis of highest relative accuracy (N.B., while maintaining a minimum separation of 5 cm between the female snout and the other snout(s)), such that the ratio between the distance of the estimated sound origin to the female snout and the male snout(s) was minimal. The reasoning was that this method could provide us with clear examples where the female did in fact vocalize. However, it is not clear at all from looking at these individual examples that the females did in fact vocalize, and rather seem to reflect either (i) extremely rare (<0.1%) cases where the assigned identity differed between the Cam64 and SLIM methods while simultaneously providing usable MPI criteria for both methods that was higher for the method where the female was closest. To be clear, in these cases, the Cam64 is likely more accurate and located between two mice that have their snouts in close proximity, leading to a relatively low MPI, while the SLIM method coincidentally is located precisely on a third, relatively distant, female. The other examples that rank high for female accuracy represent (ii) less rare (<1%) instances where the Cam64 does not produce a clear focal point and instead has many competing hotspots. We produced a short video (see Rebuttal Video 1 and the legend at the bottom of this document) where (i) represents the first instance shown, and (ii) represent the second and third instance shown, respectively.

To summarize, although we have a high confidence in the accuracy of our system, borderline false-positive cases for female vocalization are unavoidable. We thus cannot completely exclude the possibility of female vocalization, which is why we were tentative in our initial discussion of the subject in our manuscript, but we suspect that the number of female vocalization is likely still overestimated in our data, but likely even more in previously published studies1,2.

Reviewer #3 (Recommendations for the authors):The manuscript is well written. Here are a few comments that could help to clarify some points:l. 92-93 + l. 439-442: The overestimation of female vocalisations during male-female encounters in previous studies might also be related to the fact that the animals used were socially isolated for at least two weeks before the experiments (for instance in Neunuebel et al. 2015, eLife; Sangiamo et al. 2020 Nat Neurosci). This type of isolation is drastic for females and leads to an increase in social motivation, and therefore ultrasonic vocalisations emission even when interacting with females (see Ey et al. 2018 Frontiers Mol Neurosci).

Thank you for mentioning this important, potential confound. First, we would like to emphasize that we do not generally doubt that female vocalizations can be more abundant in other settings, e.g. multiple animals, after social isolation, etc, as also emphasized in the manuscript. To check the influence of social isolation in our case, we conducted a limited set of experiments with two female mice that were socially isolated for >7 days, and then dyadically interacted with two males subsequently. The isolation time is longer than in Ey et al. 2018, where you already found a significant difference in call rate. While this is of course not a representative sample and requires further study, we would like to share the preliminary results from these experiments with the reviewers: in the 4 experiments only a handful of USV were potentially from the female mouse (<10), despite a total of ~2000 USVs (mirroring results of at least one other study6). These USVs were checked manually, to be able to integrate the behavioral context into the assignment, i.e. the HyVL sub-estimates were shown on the video, and compared to the snout locations of the two mice, in addition to showing the preceding and following spectrograms. As assessed by the MPI criterion there are some USVs where the snouts are too close to draw any safe conclusions. One interesting issue that we noticed during the analysis is that occasionally the sound appears to pass underneath an animal, if the emitter is really close, but with the snout underneath the body of the other mouse, and is then most strongly visible on the other side, maybe reflecting off the platform (despite it being made from sound-absorbing foam).

In the experiments with the first female, it is noteworthy that one of the males was very active and emitted >1200 USVs in 8 minutes, while the other male (cagemate, interacting with the same female on the same day) emitted no vocalizations at all.

For the experiments with the second female, we observed the same pattern: one male was very active, emitting more than 800 USVs in the recording period, while the second male did not emit a single USV (again cagemate, interacting with the same female on the same day).

While we do not doubt the results from your work, particularly if they are in female(juvenile female) interactions as in your work, or in resident intruder interactions with an anesthetized intruder (e.g Hammerschmidt et al. PLOS One, 2012)7, which are beyond doubt evidence for female vocalizations. However, in our opinion this highlights the relevance of highly accurate localization systems for fully addressing this question in the future for all contexts of interest.

l. 231: Could the system be applied to other species? Which ones? With what types of adaptations?

There is a substantial number of species that the localization system could be applied to, essentially all animals that vocalize. The precise accuracy of localization will likely depend on the frequency range of vocalizations, with the highest accuracy possible for higher frequencies, however, still very high accuracy for lower frequencies, e.g. we can typically localize steps or scratches of a mouse, which have most energy <10 kHz. A non-exhaustive list of animals would be rats, cats, different species of insects (e.g. grasshoppers or crickets) and most bird species. For studies in the plane, i.e. on a flat surface the present acoustic localization system could be used 'as is', with the only required adaptation to retrain the spatial tracking. For studies in space, the analysis of the origin of the sound would have to be extended by a depth dimension, which would mostly increase computation time, but not introduce any fundamental changes to the localization analysis otherwise. Visual tracking in 3D should probably be done with depth cameras instead, and visual occlusion could become a bigger issue. We have removed the mentioning from this location and instead added a paragraph to the discussion containing the above information.

l. 314-315: As the tracking also depicts the tail basis, would it be possible to add a plot of the distance between the emitter's snout and the receiver's tail basis, to confirm the result that males vocalise mostly during anogenital sniffing of the female? This could be conducted at least on a subset of data.

No problem, we have added the corresponding plot as a new extended data figure to Figure 4. As expected, a large fraction of the male vocalizations are emitted, when the female mouse's abdomen is very close to the male snout (B). Conversely, the male abdomen was in a range of different relative locations to the female's snout (A). Note, however, that only dyadic interactions are shown here, because the tail marker was only tracked using the automatic tracking, which in turn was only available for dyadic interactions Automatic tracking was less accurate than manual tracking, which might help explain why a larger fraction of USVs were assigned to the female. See figure legend in the manuscript for additional interpretation of this result.